# Opening the Vocabulary of Egocentric Actions

**Dibyadip Chatterjee** [1]    **Fadime Sener** [2]    **Shugao Ma** [2]    **Angela Yao** [1]

[1]National University of Singapore    [2]Meta Reality Labs Research

{dibyadip, ayao}@comp.nus.edu.sg    {famesener, shugao}@meta.com

https://dibschat.github.io/openvocab-egoAR/

## Abstract

Human actions in egocentric videos often feature hand-object interactions composed of a verb (performed by the hand) applied to an object. Despite their extensive scaling up, egocentric datasets still face two limitations — sparsity of action compositions and a closed set of interacting objects. This paper proposes a novel open vocabulary action recognition task. Given a set of verbs and objects observed during training, the goal is to generalize the verbs to an open vocabulary of actions with seen and novel objects. To this end, we decouple the verb and object predictions via an object-agnostic *verb encoder* and a prompt-based *object encoder*. The prompting leverages CLIP representations to predict an open vocabulary of interacting objects. We create open vocabulary benchmarks on the EPIC-KITCHENS-100 and Assembly101 datasets; whereas closed-action methods fail to generalize, our proposed method is effective. In addition, our object encoder significantly outperforms existing open-vocabulary visual recognition methods in recognizing novel interacting objects.

## 1 Introduction

Egocentric or *first-person* videos captured from body-mounted cameras often feature the person manipulating objects with one or both hands. An action is thus conveniently defined as the composition of a verb performed by the hand with an interacting object, e.g., the action *"slicing apple"* is defined as *"slice"* and *"apple"*. Egocentric datasets [17, 8, 51] have scaled up by increasing the variety of verbs and objects. Theoretically, the increase in feasible actions should be quadratic, yet actually observed compositions are sparse, e.g., EPIC-KITCHENS-100 [8] has 97 verbs and 300 objects, but only 14% of possible verb-object compositions are observed as actions. Although not all compositions are feasible, e.g., *"eat dishwasher"*, many feasible actions are not observed, e.g., *"serve chicken", "crush ice"*.

We observe that the verbs in egocentric datasets tend to be domain-agnostic. For example, even though EPIC100 is set in the kitchen, its verbs like *pull*, *remove*, and *shake* are also applicable outside of the kitchen. Ideally, we want to learn the verb concepts and generalize the verb to any object. Yet the dataset bounds the type and number of objects and the capture environment, e.g., EPIC100 has cooking utensils and food, whereas Assembly101 [51] has toy vehicle parts. Simply put, the objects restrict the subsequent action space of the dataset.

Existing action recognition models [61, 11, 43, 48, 22] are designed for a closed set of actions. A few works have studied open-set action recognition [4, 65], but they simply reject actions not seen during training. Inspired by the recent success of open vocabulary object detection [63, 18, 68], a more practical objective would be to provide novel actions as text inputs during inference. To that end, we address this problem of *open vocabulary action recognition* where a video model trained on a *base* vocabulary of actions needs to recognize *novel* actions outside of that vocabulary.

We tackle open vocabulary action recognition as two problems – *(1)* generalizing verbs to novel objects and *(2)* recognizing novel objects from egocentric videos. Note that this differs from a zero-shot setup, where some actions are not observed during training. These unobserved actions are not novel because the vocabulary is known and fixed during training, hence closed. More recently, [25, 57]

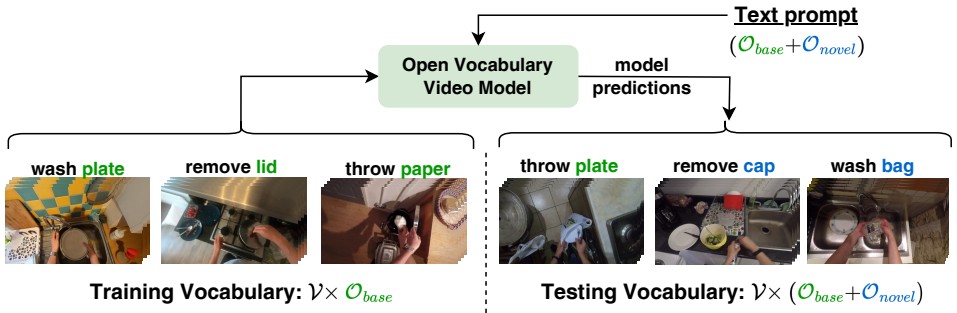

Figure 1: Open vocabulary action recognition for verb-object composed actions. During training, the model observes a predefined vocabulary of verbs and objects. During testing, the model is presented with seen verbs with any object (described by a text prompt). Objects in **green** are base classes ($\mathcal{O}_{base}$) seen during training; those in **blue** are novel classes ($\mathcal{O}_{novel}$) and unknown during training.

proposed recognizing an open vocabulary of verbs by leveraging large-scale vision-language models (VLMs) [46]. Contrary to their task, we are interested in the scenario of generalizing verbs to an open vocabulary of objects.

Generalizing verbs to novel object categories is challenging when the classes are naturally long-tailed [31, 44]. Models trained on such imbalanced datasets are biased towards head verb-object composed actions (when a verb is frequently seen with a particular object). Furthermore, these models suffer from static bias [7, 28] and overfit to the appearance of fixed frames without learning the transformation of objects over time [35]. To alleviate these biases, we decouple the verb and object predictions via two separately trained encoders. We propose an **O**bject **A**gnostic **P**retraining (OAP) for the verb encoder. OAP is based on a supervised contrastive loss [26], to which we add a set of *guiding augmentations*. These augmentations are specially designed for egocentric videos and allow the verb encoder to learn *object-agnostic* representations to improve transfer to novel objects. To our knowledge, this work is the first to explore contrastive learning of video backbones end-to-end on egocentric videos. In contrast, the augmentations in existing video contrastive learning [12, 45] are designed for Kinetics [6] styled videos where actions can be predicted from a single frame or a short clip sampled anywhere from the video. Such augmentations are ineffective for egocentric footage (Sec. C of Supplementary).

To recognize novel objects, we design an open-vocabulary object encoder that leverages a pretrained CLIP [46] model. We are specifically interested in recognizing *active* objects – those that undergo a state change from the actor's interactions. Active objects are not necessarily *handheld* [64, 9] e.g., while *"beating egg"*, the actor holds a *whisk* and a *bowl* in his hand, but the *egg* is the active object undergoing a state change. Recognizing active objects in egocentric videos is challenging even under closed settings due to the many distractor objects in natural environments. To this end, we propose **A**ctive **O**bject **P**rompting (AOP) that guides the CLIP model to understand which object is active. Given a frame containing a whisk, bowl, egg batter, hands, and other distractors, AOP aims at learning the context *i.e. "beating egg"* by generating a set of learnable verb-conditioned prompts. Conditioning the prompts on the verb features generated by OAP helps the CLIP recognize the active object.

Egocentric datasets [8, 51] focus on a closed set evaluation of actions where a handful of objects are unseen. However, for a realistic open vocabulary evaluation, a variety of novel objects is expected to be encountered for the first time during inference. To solve this, we create open vocabulary benchmarks on EPIC100 and Assembly101 by repurposing their original split. Our contributions can thus be summarized as: *(i)* an object agnostic pretraining of video backbones to improve its verb generalization to novel objects, *(ii)* an active object prompting of CLIP to recognize novel active objects and *(iii)* open vocabulary action recognition benchmarks on EPIC100 and Assembly101.

## 2   Related Works

**Egocentric Action Recognition.**   Egocentric computer vision has garnered huge interest in recent years, and massive efforts at scaling up have resulted in the development of datasets like Ego4D [17],

EPIC-KITCHENS-100 [8] and Assembly101 [51]. It has also aided the development of sophisticated video recognition architectures [11, 3, 43, 58]. Recent works [48, 22, 33] focus on learning more object-centric representations by explicitly modeling object transformations over time with off-the-shelf detectors [50] and trackers [5]. However, these models are restricted to a closed set of predefined actions and do not scale to the diversity of objects in the real world.

**Contrastive Learning in Videos.** Videos provide unique opportunities for contrastive learning [19, 41] due to its multimodal and highly redundant nature. Most popular methods follow the self-supervised instance discrimination task [59] where a single positive of an anchor video is contrasted against a set of negatives. This positive can be an augmented view of the RGB stream [12, 45, 49] or sampled from a complementary modality like optical flow [40, 60], text [14, 36, 62, 21, 30], audio [2, 27, 34, 15] or even all three at once [1, 53]. Contrary to instance discrimination, recent works [26, 20] propose supervised contrastive learning. They show that sampling multiple positives from the same class as the anchor outperforms instance discrimination. Another line of research uses multiple positives to combat noisy anchor-positive alignment [37]. In this paper, we propose a set of lightweight augmentations to train video backbones for verb prediction. These augmentations result in a noisy alignment due to the camera motion inherent in egocentric videos. Thus, we leverage multiple positives by modifying [37] to combat it.

**Learning for an Open vocabulary.** The remarkable zero-shot capabilities of large language models (LLMs) [10, 47] and vision language models (VLMs) [46, 23] have widened the possibility of recognition and detection tasks in the visual domain. Of particular interest is the open vocabulary setting, which can recognize novel classes unknown during training. By presenting the novel classes as a textual phrase, it becomes possible to probe the huge corpora of knowledge in LLMs and VLMs. It differs from the traditional zero-shot setting where the novel classes are assumed to be known during training [63] and hence is restricted to the closed vocabulary of predefined classes. Most works for open vocabulary recognition are limited to the image domain [63, 67, 66, 18, 68, 39]. Recently, some works [56, 32, 25, 57] have adapted VLMs for video tasks but primarily focus on zero-shot verb recognition. In contrast, we are interested in a practical hand-object interaction scenario of generalizing verbs performed by hands to any object.

**Prompting VLMs.** Large Scale Vision Language Models (VLMs) like CLIP [46] and ALIGN [23] are trained on an enormous corpus of web data and contain semantically grounded knowledge of images. Prompting [24] refers to designing textual instructions to steer the VLM towards some desired output. Manually engineering discrete text prompts is laborious and requires domain expertise. One alternative is fine-tuning the VLM encoder for the task at hand; this approach incurs significant computation and is also at risk of catastrophic forgetting [29].The more widely adopted approach is simply learning the prompts by backpropagating the task-specific loss while freezing the VLM encoders [67, 66, 13]. Such an approach is parameter-efficient since only a few prompt embeddings are learned while keeping the encoders frozen. In this paper, we are interested in such parameter-efficient prompt learning to steer the VLM toward active object recognition.

## 3 Preliminaries

Consider a video dataset consisting of verbs $\mathcal{V}$ applied to a base set of objects $\mathcal{O}_{base}$. The possible set of base actions is $\mathcal{A}_{base} = \mathcal{V} \times \mathcal{O}_{base} = \{(v, o) \mid v \in \mathcal{V}, o \in \mathcal{O}_{base}\}$. During training, only a sparse subset of actions are observed [1] *i.e.* $\mathcal{A}_{train} \subset \mathcal{A}_{base}$. The goal is to learn a model $f : \mathbb{X}_{train} \rightarrow \mathcal{A}_{base}$ where $\mathbb{X}_{train}$ is the set of videos seen during training. During inference, along with base objects, we also encounter novel objects $\mathcal{O}_{novel}$ that form novel actions $\mathcal{A}_{novel} = \{(v, o) \mid v \in \mathcal{V}, o \in \mathcal{O}_{novel}\}$. We want our model $f$ to predict both base and novel actions *i.e.* $\mathcal{A}_{test} = \mathcal{A}_{base} \cup \mathcal{A}_{novel}$. Conventional action recognition follows a closed vocabulary evaluation where $\mathcal{A}_{test} = \mathcal{A}_{base}$.

## 4 Method

We train two encoders – one for predicting verbs and another for predicting an open vocabulary of objects as depicted by the pink and yellow blocks in Fig. 2a. During inference, predictions from the verb and object encoders are composed to generate the action prediction.

---

[1]not all unobserved actions are feasible

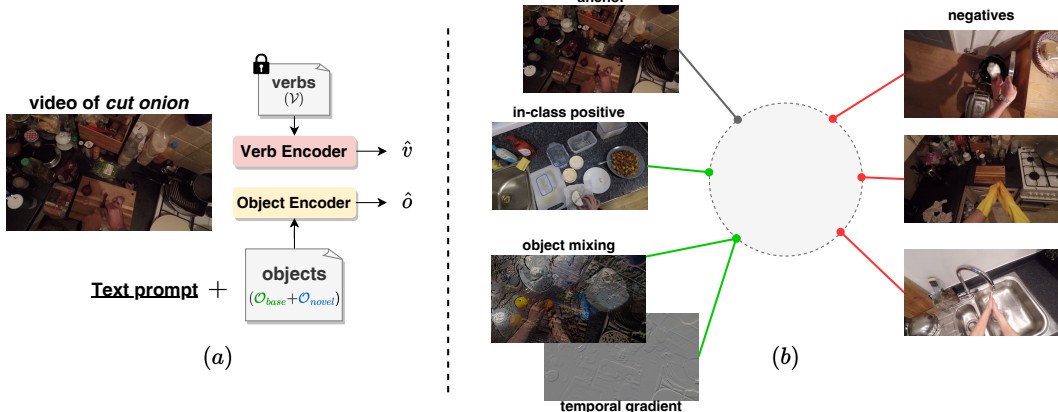

$(a)$                 $(b)$

Figure 2: $(a)$ **Framework.** We train separate encoders for predicting verb $\hat{v}$ and object $\hat{o}$. The verb encoder operates on a closed set of verbs, whereas the object encoder, trained on $\mathcal{O}_{base}$, can predict any object described by a text prompt. $(b)$ **Object Agnostic Pretraining of the Verb Encoder.** We design an object mixing augmentation to facilitate learning object-agnostic verb representations. Multiple positives are sampled for each anchor video, including our proposed object mixing (mixed with *"cut orange"*), temporal gradients of the anchor video, and in-class positives *i.e.* videos featuring the same verb (*"cut cheese"*). The two augmentations corresponding to the same point in the embedding space illustrate that the anchor only needs to minimize its distance from a softmax weighted average of the two (softmax over the similarity of the anchor with the two).

**Verb Encoder.**     Encoding verbs requires spatiotemporal modeling, so we use a video backbone as the verb encoder. We want to learn object-agnostic verb representations to encourage generalization to novel objects. We propose Object Agnostic Pretraining (OAP) (described in Sec. 4.1), which is a general pretraining strategy that can be applied to any video backbone and does not add parameters or increase the model size.

**Object Encoder.**     We aim to recognize an open vocabulary of *active objects*. Even with a closed vocabulary, finding the active object amongst the many other objects in a cluttered scene is difficult. As such, state-of-the-art open vocabulary object detectors [18, 68] perform poorly. One option for support is to spatially locate a region of interaction in the frame; we use a pretrained hand-object interaction (HOI) detector, 100DOH [52] as the base of our object encoder. It provides bounding boxes of the left and right hands and the corresponding object(s)-in-contact in a class-agnostic fashion. Note that the object bounding boxes may contain more than one object (Fig. 3). Thus, we propose an Active Object Prompting (AOP) strategy (described in Sec. 4.2) by leveraging CLIP [46] to generate active object labels from the interaction region for both base and novel classes.

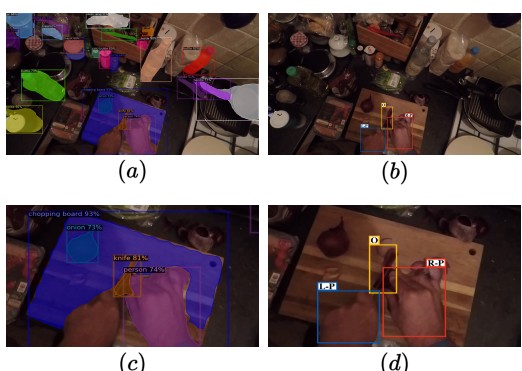

$(a)$               $(b)$

$(c)$               $(d)$

Figure 3: Example frame of *"cut onion"* from EPIC. $(a)$ DETIC [68] output with EPIC noun vocabulary. $(b)$ HOI Detector [52] output. $(c)$ & $(d)$ are crops of $(a)$ & $(b)$ respectively for better visualization. The object-in-contact crop **O** alongside *onion* also captures *knife*.

## 4.1    Object Agnostic Pretraining (OAP)

OAP aims to pull instances with the same verb as the anchor close in the embedding space (irrespective of its object class) and contrast them against instances with different verbs. We randomly sample a batch $B = \{(x_1, y_1), ..., (x_i, y_i), ..., (x_N, y_N)\}$ where $y_i$ is the verb class for the raw video clip $x_i \in \mathbb{X}_{train}$ and $N$ is the batch size. We first use

random photometric augmentations like Gaussian blur and color jittering to create two views of each anchor, thus doubling the batch size to $2N$. Each view is then fed to the verb encoder $f_\theta$ followed by a projection head $g_\theta$ which generates features for the whole batch $\{z_1, z_2, ..., z_{2N}\}$. These features are then normalized to the unit hypersphere. The verb encoder is trained with the standard supervised contrastive loss [26]. Videos sampled from $B$ with the same verb as the anchor are considered positives. For each anchor $z_i$ we have the bag of positives $\mathcal{P}_i = \{z_j | y_j = y_i, \forall j \in [1, 2N]\}$. We use the loss $\mathcal{L}_{out}$ with temperature parameter $\tau_{out}$:

$$\mathcal{L}_{out}^{(i)} = -\frac{1}{|\mathcal{P}_i|} \sum_{j \in \mathcal{P}_i} \log \frac{\exp\left(z_i \cdot z_j / \tau_{out}\right)}{\sum\limits_{k \in B \setminus i} \exp\left(z_i \cdot z_k / \tau_{out}\right)}. \tag{1}$$

**Guiding Augmentations.** Sampling in-class positives is still bounded by the verb-object compositions in the training set. Our intuition is that as long as a verb is being performed in a video, exchanging objects and the presence of other distractor objects in the background should not affect the outcome of the verb encoder. To reduce the reliance on such objects for verb prediction, we propose a set of guiding augmentations to diversify the set of positives. The augmentations are lightweight and allow new positives to be generated online during training.

To achieve this, we first mask static information (objects in the background) from a frame of the video clip. The most obvious choice is to use *temporal gradient* that eliminates any static information and focuses primarily on the region with non-zero motion vectors, which we refer to as the active region [2]. In hand-object interactions, this active region is expected to contain the active object. Furthermore, we exploit the compositional nature of actions and propose a new *object mixing* augmentation. The idea is simple – we combine two clips with the same verb but different objects to generate new object mixed clips (Fig. 2b). Formally, given two clips from the batch, $x_i$ and $x_j$, having the same verb, we get the object mixed clips $x_i^{\backprime}$ and $x_j^{\backprime}$.

$$\begin{aligned} x_i^{'} &= \alpha(\nabla x_i \odot x_i) + (1 - \alpha)[\,(\nabla x_j)^{-1} \odot x_j] \\ x_j^{'} &= \alpha(\nabla x_j \odot x_j) + (1 - \alpha)[\,(\nabla x_i)^{-1} \odot x_i] \end{aligned} \tag{2}$$

where $\nabla x_i$ is the first-order temporal gradient of $x_i$ (which masks static regions) and $\odot$ is the element-wise matrix multiplication; $(\nabla x_j)^{-1}$ refers to the inverted gradient image of $x_j$ and $\alpha$ is the mixing hyperparameter. To summarize, masking aims at minimizing the impact of distractor objects outside of the active region, and mixing prevents the model from being biased towards low-level object information, e.g., color, shape, and hence to the observed verb-object compositions.

**Learning from *Noisy* Augmentations.** The basis of our guiding augmentations is the temporal gradient. In egocentric videos, the camera (ego) motion will result in non-zero gradients on static parts of the scene in addition to the true active region. While it is feasible to specially design expensive augmentations that account for camera motion [55, 54]; instead, we propose to adjust the contrastive loss so that it can learn from these noisy but lightweight augmentations. In $\mathcal{L}_{out}$, all the positives for an anchor contribute equally to the loss; this is desirable for in-class positives where the condition for being a positive is well-defined and fixed, but undesirable for the noisy guiding augmentations. Now, if we move the summation inside the *log*, the *logit* for each anchor becomes the average cosine similarity of *all* the guiding augmentations. The contribution of each augmentation is thus dependent on its similarity with the anchor, where noisy augmentations having low cosine similarity will have less contribution to the gradient. With a sufficiently large bag of augmentations, this is expected to reduce the noise implicitly. Such a loss is formulated as $\mathcal{L}_{in}$ in [26]. If $\mathcal{G}_i$ is our bag of guiding augmentations, our $\mathcal{L}_{in}$ loss is as follows:

$$\mathcal{L}_{in}^{(i)} = -\log \frac{\frac{1}{|\mathcal{G}_i|} \sum\limits_{j \in \mathcal{G}_i} \exp\left(z_i \cdot z_j / \tau_{in}\right)}{\sum\limits_{k \in B \setminus i} \exp\left(z_i \cdot z_k / \tau_{in}\right)} \tag{3}$$

where $\tau_{in}$ is the temperature parameter. This loss is also similar to MIL-NCE [37] for $\tau_{in} = 1$, which is used for learning from noisy video-text pairs. In our case, a lower softmax temperature $(< 1)$

---

[2]active regions may still contain static objects, e.g., for the action of "cutting onions", the cutting board which is present in the active region is static. This object is not a distractor and is relevant for action prediction.

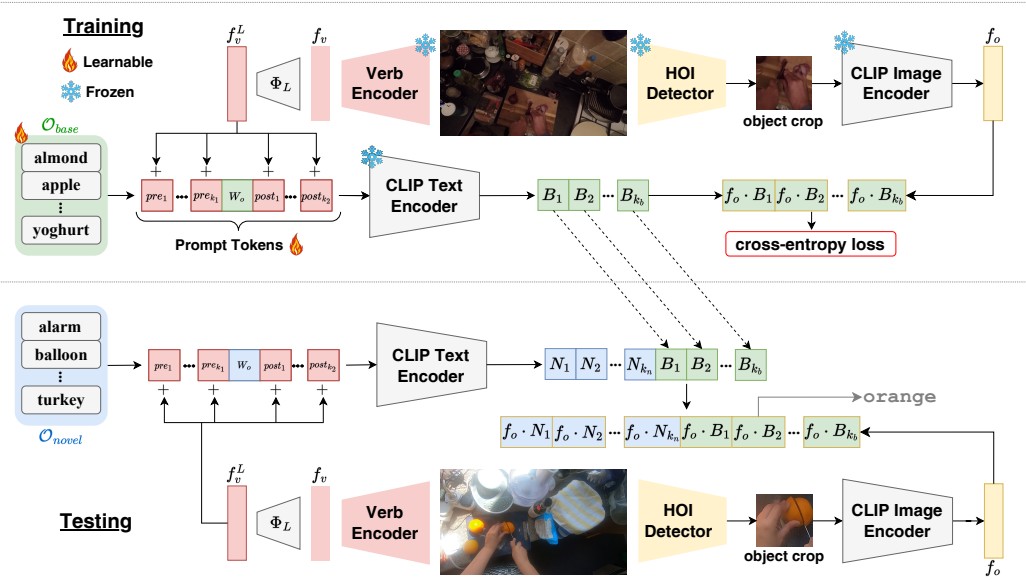

Figure 4: **(Top) Parameter-efficient training for Active Object Prompting.** Keeping the CLIP encoders frozen, the base vocabulary initialized with the CLIP-pretrained word embeddings is fine-tuned on the training set. A set of prefix and postfix prompt vectors conditioned on the verb representation are also learned. **(Bottom) Testing with Active Object Prompting**. The fine-tuned embeddings are used for the base objects, whereas the CLIP-pretrained embeddings are used for novel ones. The learned prefix and postfix prompts are shared among all object classes.

results in better performance. Please refer to the Supplementary for theoretical insights into the loss formulation (Sec. D) and illustrations of guiding augmentations (Sec. F). Thus, the overall loss function is $\mathcal{L} = \mathcal{L}_{out} + \lambda \mathcal{L}_{in}$, where $\lambda$ is a weighting hyperparameter.

## 4.2 Active Object Prompting (AOP)

This section discusses leveraging CLIP to predict the active object from a crop with multiple objects. We assume no prior knowledge of the set of objects observed during inference *i.e.* any object is expected to be predicted by our model as long as it is described by a textual phrase (open vocabulary).

The overall framework of AOP is shown in Fig. 4. First, the given set of object classes $\mathcal{O}$ are converted into text prompts by prepending and appending each class with a set of prefix and suffix tokens. For object $o_i$ with word embedding $W_{o_i}$, a prompt text $t_i$ can be defined as $t_i = \{\text{pre}_1, ..., \text{pre}_{k_1}, W_{o_i}, \text{post}_1, ..., \text{post}_{k_2}\}$, where $\text{pre}_j$ are $k_1$ prefix tokens, $\text{post}_j$ are $k_2$ postfix tokens. The prefix and postfix prompts, together known as context prompts [67], are essential in guiding a pretrained CLIP to solve a desired task. The context prompt tokens are shared across all classes and can be fixed or learned.

Fixed prompt tokens must be manually engineered by humans, which is laborious and requires domain expertise. Also, prompting CLIP, which operates in a continuous space with discrete human-readable text prompts, limits its usage. Hence, we propose to learn our prompts via two prompting strategies – *(i)* verb-conditioned context prompt generation and *(ii)* fine-tuning base object vocabulary.

**Verb-Conditioned Context Prompt Generation.** As shown in Fig. 3, given a crop containing onion, knife cutting board and hands, we want the CLIP to predict onion as the active object by understanding the context *i.e.* the action of *cut onion*. To this end, we propose verb-conditioned context prompts. Given the video $x$, we add the verb representation of $x$ learned using OAP to the context prompts as shown in Fig. 4 *i.e.* $\text{pre}_j = \text{pre}_j + \Phi_L(f_\theta(x))$ and $\text{post}_j = \text{post}_j + \Phi_L(f_\theta(x))$, where $f_\theta$ is the pretrained verb encoder and $\Phi_L$ is a projection network used to project the visual feature to the language domain.

**Fine-tuning the Base Object Vocabulary.** Egocentric videos often capture many different objects in the same environment. Semantic variability of the objects can be very fine-grained e.g. *"boom"* vs. *"arm"* in Assembly101 [51]. Such fine granularity is not fully captured by CLIP pretrained on data crawled from the web. We treat this as a vocabulary learning problem with the goal of making fine-grained object categories discriminative in the continuous CLIP feature space. To this end, we fine-tune the pretrained object embeddings $W_{o_i}$ for all base objects seen during training *i.e.* $\forall o_i \in \mathcal{O}_{base}$. This simple fine-tuning allows CLIP to adjust the object embeddings in its continuous space to make them discriminative for optimal performance.

The CLIP-text encoder is kept frozen for both the prompting strategies, making them parameter-efficient. During testing (Fig. 4 bottom), we use the fine-tuned embeddings for the base objects and the original pretrained embeddings for the novel objects. The verb-conditioned context prompts are unique for every instance.

# 5 Experiments

**Datasets.** We conduct experiments on three datasets: **EPIC100** [8] features 100 hours of ego-centric footage of daily kitchen activities annotated with 97 verbs and 300 interacting objects. **Assembly101** [51] is a multi-view procedural activity dataset featuring 4321 videos of people assembling and disassembling 101 take-apart toy vehicles and is annotated with 24 verbs and 90 objects. We use the egocentric view *e3* (see [51]), which best captures the hands and interacting objects. **Something-Else** [35], is a human-object interaction subset of SSv2 [16] with 174 verb phrases *e.g. "picking something up"*. Something-Else is not strictly egocentric but shares interesting properties such as diverse hand-object interactions and active objects undergoing state change. We use the compositional split, where all verb phrases are seen during training but applied to novel objects during testing. Interacting objects per video are not annotated; hence, it is unsuitable for our open vocabulary action recognition task but is an ideal testbed for evaluating our verb encoder and OAP with other baselines.

**Implementation Details.** We use S3D [61] as the verb encoder for the majority of our experiments and ablations as it is a popular architecture for video contrastive learning [20, 37]. For all datasets, the verb encoder is pretrained with the OAP contrastive scheme for 300 epochs and then fine-tuned for verb classification for another 100 epochs. To see how OAP scales with larger backbones, we also evaluate Mformer-B [43] on Something-Else. For the hand-object interaction support, we use the pretrained 100DOH [52] detector without fine-tuning on the target dataset. For the CLIP Image Encoder, we experiment with ViT-B/16 and the larger ViT-L/14 backbones. Keeping both the CLIP image and text encoders frozen, prompts are learned using AOP for 20 epochs. Unless otherwise stated, we choose $k_1$ and $k_2$, the number of learnable prefix and postfix prompts, to be 16 each. For additional implementation details, please refer to Sec. B of the Supplementary.

Table 1: Proposed open vocabulary benchmark splits.

| Dataset | $\mathcal{V}$ | $\mathcal{O}_{base}$ | $\mathcal{O}_{novel}$ | #train seg | #test seg. |
|---|---|---|---|---|---|
| EPIC100-OV | 97 | 190 | 110 | 63k | 13.8k |
| Assembly101-OV | 24 | 60 | 30 | 56.2k | 15k |

## 5.1 Open Vocabulary Benchmark

Since the original action recognition split of EPIC100 and Assembly101 are unsuitable for open vocabulary (OV) evaluation, we create OV benchmarks EPIC100-OV and Assembly101-OV, respectively, by dividing the complete set of objects into base and novel categories. The base objects are seen during training, whereas the novel objects are kept only for testing. We also incorporate base actions in our test set to evaluate the closed-world performance of the models. The benchmark statistics are provided in Tab. 1. We provide a more detailed explanation of the OV splits in Sec. A of the Supplementary. We use Top-1 Accuracy [8, 51] as our evaluation metric and report results for both closed and novel classes. Additionally, we report the harmonic mean (HM) [38] of the closed and novel accuracies as a consolidated metric for comparing different methods.

Table 2: Compositional Action Recognition on Something-Else [35]. Extra Module refers to if any additional parameterized module has been used on top of the video backbone.

| Model | Extra Module | Top-1 (%) | Top-5 (%) |
|---|---|---|---|
| I3D [6] | ✗ | 46.8 | 72.2 |
| Slowfast [11] | ✗ | 52.2 | 80.3 |
| Slowfast+IRN [33] | ✓ | 52.9 | 80.8 |
| STRG,STIN+OIE+NL [35] | ✓ | 56.2 | 81.3 |
| S3D [61] | ✗ | 49.3 | 77.3 |
| **OAP (S3D)** | ✗ | **53.7** | **80.5** |
| Mformer-B [43] | ✗ | 60.2 | 85.8 |
| **OAP (Mformer-B)** | ✗ | **61.8** | **86.2** |

Table 3: Comparison of methods for open-vocab object recognition on EPIC100-OV. † Zero-shot evaluation.

| Model | Object Top-1 (%) | | |
|---|---|---|---|
| | Base | Novel | HM |
| S3D[†] | **52.2** | 1.6 | 3.1 |
| CLIP (ViT-B/16)[†] | 20.2 | 0.6 | 1.2 |
| CLIP (ViT-B/16) | 7.9 | 16.3 | 10.6 |
| DETIC [68] | 12.0 | 2.3 | 3.9 |
| C2 (ViT-B/16) [25] | 19.1 | 9.9 | 10.3 |
| C4 (ViT-B/16) [25] | 39.9 | 7.3 | 12.3 |
| **AOP (CLIP ViT-B/16)** | 41.9 | 19.2 | 26.3 |
| **AOP (CLIP ViT-L/14)** | 47.8 | **22.6** | **30.7** |

## 5.2 Object-Agnostic Pretraining (OAP) vs State-of-the-Art

How effective is OAP in generalizing verbs to novel objects? Tab. 2 shows OAP applied to video backbones S3D [61] and Mformer-B [43] on the Something-Else compositional split. S3D with OAP outperforms Slowfast [11] by $1.5\%$ in Top-1 acc. even though Slowfast is a stronger video backbone. It also outperforms, by $0.8\%$, the sophisticated Slowfast+IRN [33], which adds an extra transformer to SlowFast to explicitly model the hand-object interactions. Finally, S3D with OAP is still competitive with respect to STRG,STIN+OIE+NL [35], which is an ensemble of various video models, despite being a single backbone without any extra parameters or compute. Mformer-B is a state-of-the-art video architecture comfortably outperforming all other models; adding OAP improves its Top-1 performance by $1.6\%$, demonstrating that our pretraining is robust to the scale of the backbone.

## 5.3 Active Object Prompting (AOP) vs. State-of-the-Art

Tab. 3 compares AOP with similar open vocabulary baselines. The first two rows with † refer to zero-shot baselines of S3D and CLIP, where training is done using cross-entropy over all classes (base+novel). CLIP[†] refers to the linear probing of CLIP's frozen visual encoder. Both models overfit to the base classes and fail to generalize to novel ones. DETIC [68], a state-of-the-art open vocabulary object detector, improves novel object performance over S3D marginally by $1.3\%$. It is non-trivial for DETIC to recognize which object is active from a set of per-frame object detections *e.g.* for the action *"beating egg"*, a whole egg is not present in any frame. C4 [25] adapts CLIP for video tasks by adding a transformer layer on top of per-frame CLIP features; again, it overfits to the base classes. Finally, our AOP using both CLIP backbones significantly outperforms all other baselines – AOP improves over the closest competitor (C4) having the same backbone in ViT-B/16 by $14.0\%$ in HM.

## 5.4 Open Vocabulary Action Recognition

Tab. 4 presents action recognition on open vocabulary splits of EPIC and Assembly101. S3D predicts verbs and objects using two heads but share the same backbone. For a fair comparison with our decoupled verb and object encoders, we also decouple the S3D training, one for predicting verbs and another for objects (reported as $2\times$S3D). As mentioned in Sec. 5.3, the S3D baseline predicts novel objects in a zero-shot fashion where the novel categories are already assumed to be known during training. Both S3D baselines fail to predict novel objects on both benchmarks. OAP improves verb generalization to novel objects by $1.3\%$ and $4.8\%$ over S3D on EPIC100-OV and Assembly101-OV, respectively. Additionally, the closed-world verb prediction is also enhanced by $1.6\%$ and $7.0\%$ over S3D on EPIC100-OV and Assembly101-OV, respectively. AOP significantly improves novel object prediction performance in both benchmarks. The novel object prediction performance on Assembly101 could be further enhanced by adopting an HOI detector that is robust to domain shifts in the target dataset since Assembly101 egocentric videos are monochrome.

Table 4: Open Vocabulary Action Recognition Results.

| Dataset | Model | Closed Top-1 (%) | | | Novel Top-1 (%) | | | HM Top-1 (%) | | |
|---|---|---|---|---|---|---|---|---|---|---|
| | | Verb | Object | Action | Verb | Object | Action | Verb | Object | Action |
| EPIC100-OV | S3D | 62.5 | 50.8 | 37.6 | 40.1 | 1.7 | 0 | 48.8 | 3.3 | 0 |
| | 2×S3D | 61.5 | 52.2 | 36.7 | 37.9 | 1.6 | 0.1 | 46.9 | 3.1 | 0.2 |
| | **OAP + AOP** | **64.1** | 47.8 | 35.9 | **41.4** | **22.6** | **11.2** | **50.3** | **30.7** | **17.0** |
| Assembly101-OV | S3D | 50.6 | 34.6 | 19.9 | 40.3 | 0 | 0 | 44.9 | 0 | 0 |
| | 2×S3D | 50.7 | 37.9 | 20.2 | 41.8 | 0 | 0 | 45.8 | 0 | 0 |
| | **OAP + AOP** | **57.6** | 30.1 | 21.6 | **45.1** | **6.7** | **3.7** | **50.5** | **11.0** | **6.6** |

Table 5: Effect of guiding augmentations for **OAP** on Something-Else.

| Loss | Temporal Gradients | Object Mixing | Top-1 (%) | Top-5 (%) |
|---|---|---|---|---|
| Supervised | ✗ | ✗ | 49.3 | 77.3 |
| $\mathcal{L}_{out}$ | ✗ | ✗ | 50.1 | 78.1 |
| $\mathcal{L}_{out}$ | ✓ | ✓ | 52.1 | 79.3 |
| $\mathcal{L}_{out} + \lambda\mathcal{L}_{in}$ | ✓ | ✗ | 50.3 | 78.4 |
| $\mathcal{L}_{out} + \lambda\mathcal{L}_{in}$ | ✓ | ✓ | **53.7** | **80.5** |

## 5.5 Ablations

**Guiding Augmentations.** We study the effect of using guiding augmentations on Something-Else in Table 5. Using $L_{out}$ already improves performance over supervised training, consistent with the findings in supervised contrastive learning [26, 20]. Introducing verb-specific guidance via temporal gradients and especially our proposed object mixing augmentations boosts the performance further by 3.6%. This demonstrates that OAP with guiding augmentations enhances the generalization of a backbone like S3D without requiring additional parameters.

**Active Object Prompting.** We thoroughly ablate various learnable prompting strategies in Tab. 6 on EPIC100-OV. ViT-B/16 has been used as the image encoder for all experiments here. For the rows where the context prompts are not learnable, we use the fixed prompt *"A picture of a <object>."* . Introducing the HOI detector to generate active object crops improves HM by 3.1% over using full images. Our AOP mainly consists of two proposals – fine-tuning base object vocabulary and conditioning context prompts on verb representations from the video. Most of the gain in Tab. 6 comes from these two proposals. Fine-tuning $\mathcal{O}_{base}$ is expected to improve closed set performance, but we also find 2% improvement over novel categories. This alludes to the fact that learning object names decreases false positives when tasked to predict a novel category. Verb-conditioned prompting brings further improvement of 3.9% in HM. In particular, it boosts the performance over base categories, which is consistent with the findings of conditional prompting learning [66, 67].

Next, we analyze using temporal modeling on top of the CLIP features. Similar to C4 [25], we add a single-layer transformer with eight heads over the per-frame CLIP features. As expected, the model overfits to the base categories seen during training, only achieving 4.8% over novel categories.

Table 6: **AOP ablations on EPIC100-OV.** ViT-B/16 is used as the image encoder. *AOP-Base* and *AOP-Novel* refer to the best-performing models on base and novel object categories.

| | HOI Detector | Learnable Context Prompts | Verb Conditioned | Finetune $\mathcal{O}_{base}$ | Temporal modelling | Base | Novel | HM |
|---|---|---|---|---|---|---|---|---|
| | ✗ | ✗ | ✗ | ✗ | ✗ | 5.4 | 12.1 | 7.5 |
| | ✓ | ✗ | ✗ | ✗ | ✗ | 7.9 | 16.3 | 10.6 |
| | ✓ | ✗ | ✗ | ✓ | ✗ | 20.2 | 18.3 | 19.2 |
| | ✓ | ✓ | ✗ | ✓ | ✗ | 21.6 | 16.6 | 18.8 |
| *AOP-Novel* | ✓ | ✓ | ✓ | ✓ | ✗ | 29.6 | 18.4 | 22.7 |
| | ✓ | ✓ | ✗ | ✗ | ✓ | 46.9 | 4.8 | 8.7 |
| *AOP-Base* | ✓ | ✓ | ✓ | ✓ | ✓ | **51.0** | 11.9 | 19.3 |
| *AOP-Base + AOP-Novel* | ✓ | ✓ | ✓ | ✓ | ✓ | 41.9 | **19.2** | **26.3** |

Finally, we ensemble the predictions of the best-performing model on base (*AOP-Base*) and novel (*AOP-Novel*) categories to obtain the best HM.

## 5.6 Limitations and Outlook

In this paper, we assume that every verb-object composition is feasible. The compositions could be pruned to a feasible subset to reduce the action search space. Obtaining feasibilities, and distinguishing them from common-sense settings, however, is non-trivial, e.g., *"cut thermometer"*, albeit feasible, is unlikely to be observed. Our verb-conditioned prompting of CLIP provides a weak feasibility check. However, CLIP is well known to have a strong object bias [42] over verbs, and it may still yield infeasible active object predictions. One direction to improve the feasibility of actions is to learn object affordances; we leave this for future work.

## 6 Conclusion

This paper proposed a new task of open vocabulary action recognition for verb-object composed actions and a novel approach to address it. We decouple the verb and object predictions. The verb encoder is trained with an object-agnostic pretraining strategy to improve its generalization to novel objects. Our object encoder leverages a pretrained CLIP model using active object prompting. This guides the CLIP to predict an open vocabulary of interacting objects. Our experiments demonstrate that the proposed approach can recognize hand-object interactions even when the specific actions or objects are not in the training data.

**Acknowledgements:** This research/project is supported by the National Research Foundation, Singapore and DSO National Laboratories under its AI Singapore Programme (AISG Award No: AISG2-RP-2020-016). Any opinions, findings and conclusions or recommendations expressed in this material are those of the author(s) and do not reflect the views of National Research Foundation, Singapore.

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
