# Supplementary: Opening the Vocabulary of Egocentric Actions

**Dibyadip Chatterjee** [1]    **Fadime Sener** [2]    **Shugao Ma** [2]    **Angela Yao** [1]

[1]National University of Singapore    [2]Meta Reality Labs Research

{dibyadip, ayao}@comp.nus.edu.sg    {famesener, shugao}@meta.com

https://dibschat.github.io/openvocab-egoAR/

The supplementary provides additional information on our open vocabulary benchmarks (Sec. A), additional implementation details (Sec. B), spatiotemporal augmentations used for OAP (Sec. C), theoretical insights into OAP loss formulation (Sec. D), additional ablations (Sec. E) and visualizations of our proposed guiding augmentations (Sec. F).

## A    Additional Open Vocabulary Benchmark Details

The original action recognition benchmarks of EPIC-KITCHENS-100 [4] and Assembly101[23] were designed for a closed set evaluation of actions. Only a handful of objects were hidden from the training set *i.e.* only $11/300$ and $5/90$ objects were not seen during training for EPIC100 and Assembly101, respectively. As a result, we repurpose the original splits of EPIC100 and Assembly101 to create open vocabulary benchmarks EPIC100-OV and Assembly101-OV while maintaining three additional constraints – *(i)* It is natural to expect that video models, even if being evaluated on egocentric video datasets, have been pretrained on large-scale datasets like ImageNet-1k [5] and Kinetics [2]. Recently released and the largest available egocentric dataset to date, Ego4D [8] is also being used to pretrain models for various egocentric video tasks [15, 26]. To remove any overlap of objects seen in these three datasets, we purposefully keep them in the base split. *(ii)* In a real-world evaluation, novel object categories are expected to come from the rare (tail) classes [30]. As a result, we keep objects having less than $k$ instances in the novel split where $k$ is determined by the frequency distribution of actions for a dataset. *(iii)* Some of the base actions were included in the test split to evaluate performance on both base and novel categories. Following are our open vocabulary benchmarks:

Table 1: Detailed open vocabulary benchmark splits.

| Dataset | $\mathcal{V}$ | $\mathcal{O}_{base}$ | $\mathcal{O}_{novel}$ | #training | | | | #testing | | | |
|---|---|---|---|---|---|---|---|---|---|---|---|
| | | | | verbs | objects | actions | segments | verbs | objects | actions | segments |
| EPIC100-OV | 97 | 190 | 110 | 97 | 190 | 2064 | 63072 | 94 | 283 | 2639 | 13813 |
| Assembly101-OV | 24 | 60 | 30 | 24 | 60 | 871 | 56222 | 24 | 88 | 982 | 15072 |

**EPIC100-OV.**    The original EPIC100 comprises 76885 labeled segments featuring 97 verbs and 300 objects. We first divide the set of objects into base $\mathcal{O}_{base}$ and novel $\mathcal{O}_{novel}$ categories. EPIC100 has 36 objects in common with Imagenet, 46 objects in common with Kinetics and 179 objects in common with Ego4D. Therefore, 189 objects can already be seen via Imagenet/Kinetics/Ego4D pretaining, all of which are kept in $\mathcal{O}_{base}$. After this, all objects having less than ten instances are kept in $O_{novel}$. The rest are randomly assigned to roughly generate an 80/20 train/test split while ensuring all 97 verbs are seen at least once during training. Detailed statistics of the train-test split are available in Tab. 1.

**Assembly101-OV.**    The original Assembly101 comprises 71294 labeled segments featuring 24 verbs and 90 objects. Assembly101 has seven objects in common with Imagenet, 0 objects in common with Kinetics and 23 objects in common with Ego4D. Combining them, 25 objects can already be seen via Imagenet/Kinetics/Ego4D pretraining, all of which are kept in $\mathcal{O}_{base}$. Unlike EPIC100, the number

37th Conference on Neural Information Processing Systems (NeurIPS 2023).

of common objects is low, alluding to the fact that objects in Assembly101 are very domain-specific *i.e.* the task of assembling a toy vehicle. After this, all objects having less than 50 instances are kept in $O_{novel}$. The rest are randomly assigned to roughly generate an 80/20 train/test split while ensuring all 24 verbs are seen at least once during training.

## B   Additional Implementation Details

All models are implemented in PyTorch [19] with a maximum of 4 NVIDIA RTX 8000 GPUs at any time during training. We use a 16-frame clip as input for all the experiments.

**Supervised Baselines.**   For all supervised baselines (verb/object) on the respective datasets, S3D [29] is trained from a Kinetics-400 pretrained model with the standard cross-entropy loss for 200 epochs. Adam [13] is utilized for optimization, keeping a base learning rate of 0.001 with a 10 epoch warmup [7] and is decayed by a factor of 10 twice after 100 and 120 epochs, respectively. We use a 16-frame clip as input for all the supervised baselines where the frame sampling strategy described in [20] is followed for Something-Else and EPIC100. For Assembly101, the sampling strategy described in the original paper [23] is followed. During inference, following standard practice in action recognition [28, 14], we sample multiple clips and average their predictions.

**Object Agnostic Pretraining.**   S3D, initialized from a Kinetics-400 checkpoint, is pretrained using OAP on the respective datasets for 300 epochs. Hyperparameters $\alpha$ and $\lambda$ are chosen to be 0.5 and 1, respectively. For optimization, Adam is used with a base learning rate of 0.001 with a 10 epoch warmup and cosine scheduling. For contrastive learning, we use a momentum encoder [10, 3] with a memory queue of size 8192 to cache the negatives and in-class positives. The guiding augmentations are calculated on the fly for each instance and are not cached into the memory queue to maintain the diversity. $\tau_{out}$ is set as 0.07 (MoCo default), and an ablation of $\tau_{in}$ is provided in Sec. D. For compositional action recognition on Something-Else [18], Mformer-B initialized from a Kinetics-600 checkpoint, is pretrained using OAP with AdamW [16] optimizer and a base learning rate of 0.0001 keeping all other hyperparameters similar to S3D. OAP pretrained S3D and Mformer-B are fine-tuned for verb classification for 100 and 35 epochs, respectively.

**Active Object Prompting.**   We use 100DOH [24] as our hand-object detector (not fine-tuned on the target dataset due to lack of hand-object bounding box annotations). We use CLIP [22] as our pretrained VLM; especially, we use ViT-B/16 and ViT-L/14 as backbones of the CLIP image encoder. Both CLIP image and text encoders are frozen for all experiments and only the base object vocabulary or the prefix/postfix prompts are learned. We vary the number of learnable prefix and postfix prompts and find 16 each, resulting in the best overall performance (HM). We also find that learning multiple tokens for each class in the base vocabulary outperforms learning one token per class (Sec. E.3). For verb-conditioned prompting, we follow the meta-network design from [31] *i.e.* a 2-layer bottleneck MLP (Linear-ReLU-Linear) where the hidden dimension is reduced by $16\times$. These prompts are learned using Adam optimizer for 20 epochs with a base learning rate of $1e-04$, 1 warmup epoch, and weight decay of $1e-05$. By default, predictions from 16 frames (the same frames that were input to the verb encoder) are averaged. For the experiments with temporal modeling, following [11], a single transformer layer with 8 heads is applied over per-frame CLIP features.

**Ensembling for AOP.**   We take the two best-performing models on the base and novel object categories, namely *AOP-Base* and *AOP-Novel* respectively, and ensemble their predictions to achieve a better balance of performance in both. As seen from Tab. **??**, such ensembling results in the best overall HM. We use a weighted arithmetic mean of predictions from the two models:

$$p_{ensemble}^c = \begin{cases} \gamma \cdot p_{AOP\text{-}Base}^c + (1-\gamma) \cdot p_{AOP\text{-}Novel}^c & \text{if } c \in \mathcal{O}_{\text{base}} \\ (1-\gamma) \cdot p_{AOP\text{-}Base}^c + \gamma \cdot p_{AOP\text{-}Novel}^c & \text{if } c \in \mathcal{O}_{\text{novel}} \end{cases} \tag{1}$$

where $\gamma$ is set to be 0.56. This formulation is similar to [9], but instead of geometric mean, we find arithmetic mean to perform better in practice.

## C    Spatiotemporal augmentations for OAP

Existing video contrastive learning methods are almost always pretrained on Kinetics [2] and designing random spatiotemporal augmentations for such cases is well-summarized in [6, 21].

**Spatial Augmentations.**    Widely used spatial augmentations include random cropping, horizontal flipping, color jittering and Gaussian blur. CVRL [21] proposes to use temporally consistent spatial augmentations *i.e.*having a fixed randomness across the frames of a video. We follow their temporally consistent design for random color jittering, gaussian blurring and horizontal flipping. However, we refrain from using horizontal flipping for Something-Else because the presence of actions like *"pushing something from left to right"* with our supervised contrastive loss will result in false positives. We also avoid random cropping since it often results in cropping out hands and objects from the frame [17]. Actions in SSv2 have a center bias, while actions in EPIC100 and Assembly101 have a bottom corner bias. Thus, we use center cropping for Something-Else and bottom vertical cropping (crop starting from the bottom up to height $H$ while preserving the entire width) for EPIC100 and Assembly101.

**Temporal Augmentations.**    The majority of Kinetics actions are repetitive (*e.g. brushing, clapping*) or have a strong scene bias (*e.g. bowling, surfing*) such that it can be recognized from a single frame or a short clip sampled anywhere from the video. Temporal augmentations designed for Kinetics thus include randomly sampling short clips from different timestamps of the video. In contrast, Something-Else and egocentric videos feature highly temporal *durative* actions *i.e.* actions occur over the full duration of the video. Randomly sampling clips from durative actions will lead to learning temporally agnostic features. We observe that the point-of-no-return (PNR [1]) frame generally occurs around the middle of the video. Hence, in order to sample $n$ frames from a video having $T$ frames, we first randomly sample start and end frames from $[0, \frac{T-n}{2}]$ and $[\frac{T+n}{2}, T]$ respectively and then uniformly sample $n$ frames from the start to end interval.

## D    Theoretical insights into $\mathcal{L}_{out}$ vs $\mathcal{L}_{in}$

For an anchor video $x_i$ and its normalized feature projection $z_i$, our $\mathcal{L}_{out}$ and $\mathcal{L}_{in}$ losses are:

$$\mathcal{L}_{out}^{(i)} = -\frac{1}{|\mathcal{P}_i|} \sum_{j \in \mathcal{P}_i} \log \frac{\exp\left(z_i \cdot z_j / \tau_{out}\right)}{\sum\limits_{k \in B \setminus i} \exp\left(z_i \cdot z_k / \tau_{out}\right)} \tag{2a}$$

$$\mathcal{L}_{in}^{(i)} = -\log \frac{\frac{1}{|\mathcal{G}_i|} \sum\limits_{j \in \mathcal{G}_i} \exp\left(z_i \cdot z_j / \tau_{in}\right)}{\sum\limits_{k \in B \setminus i} \exp\left(z_i \cdot z_k / \tau_{in}\right)} \tag{2b}$$

where $\tau_{out}$ and $\tau_{in}$ are temperature parameters and $\mathcal{G}_i$ is the bag of guiding augmentations. We set $\tau_{out}$ to MoCo default 0.07 and ablate $\tau_{in}$ in Tab. 2. We observe that lower temperatures $< 1$ perform better and the performance degrades with higher temperatures. MIL-NCE [1], a popular method to align noisy positives, uses the same loss formulation as $\mathcal{L}_{in}$ without the $\frac{1}{|\mathcal{G}_i|}$ term and $\tau_{in} = 1$.

Table 2: $\tau_{in}$ **ablation on Something-Else for** $|\mathcal{G}_i| = 5$

| $\tau_{in}$ | 0.01 | **0.1** | 0.5 | 1.0 | 10.0 |
|---|---|---|---|---|---|
| Top-1 (%) | 53.5 | **53.7** | 53.4 | 52.6 | 52.3 |

Now, in order to understand the effect of $\tau_{in}$ and our intuition behind using $\mathcal{L}_{in}$ instead of $\mathcal{L}_{out}$ for the noisy guiding augmentations, we compute the gradient of the losses with respect to $z_i$. Following [12], this can be simplified into the following forms:

---

[1]Ego4D [8] defines PNR frame as the beginning of an object state change

$$\frac{\partial \mathcal{L}_{out}^{(i)}}{\partial z_i} = \frac{1}{\tau_{out}} \left\{ \underbrace{\sum_{p \in \mathcal{P}_i} z_p \left( P_{ip} - \frac{1}{|\mathcal{P}_i|} \right)}_{\text{gradients from positives}} + \underbrace{\sum_{n \in \mathcal{N}_i} z_n P_{in}}_{\text{gradients from negatives}} \right\} \tag{3a}$$

$$\frac{\partial \mathcal{L}_{in}^{(i)}}{\partial z_i} = \frac{1}{\tau_{in}} \left\{ \underbrace{\sum_{p \in \mathcal{G}_i} z_p \left( P_{ip} - \frac{\exp\left(z_i \cdot z_p / \tau_{in}\right)}{\sum\limits_{p' \in \mathcal{G}_i} \exp\left(z_i \cdot z_{p'} / \tau_{in}\right)} \right)}_{\text{gradients from positives}} + \underbrace{\sum_{n \in \mathcal{N}_i} z_n P_{in}}_{\text{gradients from negatives}} \right\} \tag{3b}$$

where $P_{ip} \equiv \frac{\exp(z_i \cdot z_p / \tau)}{\sum\limits_{k \in B \setminus i} \exp(z_i \cdot z_k / \tau)}$ is the likelihood of $p \in \mathcal{P}_i$ with respect to all elements in the batch and $\mathcal{N}_i = \{z_j | y_j \neq y_i, \forall j \in [1, \ 2N]\}$ is the set of negatives for anchor $x_i$ in the batch. We are interested in gradients from the positives. From Eq. 3a, it can be observed that for $\mathcal{L}_{out}$, all positives contribute equally to the gradient since a constant term $\frac{1}{|\mathcal{P}_i|}$ is subtracted from the likelihood $P_{ip}$. Whereas in Eq. 3b, $\mathcal{L}_{in}$ implicitly weights each positive (guiding augmentation) via subtracting from $P_{ip}$, the likelihood of a positive ($p$) with respect to all positives ($p' \in \mathcal{G}_i$). Increasing $\tau_{in}$ will soften the weighting term and at the very extreme, for a very high $\tau_{in}$, the weighting term will degenerate to $\frac{1}{|\mathcal{P}_i|}$, thus making $\mathcal{L}_{in} \equiv \mathcal{L}_{out}$. This suggests increasing the temperature will make all guiding augmentations contribute equally to the gradient, which we are trying to avoid due to the noisy nature of some positives. This conclusion is also empirically supported in Tab 2. We expect given a sufficiently large bag of augmentations (Sec. E.1), $\mathcal{L}_{in}$ can provide valuable object-agnostic information to regularize $\mathcal{L}_{out}$.

# E  Additional Ablations

In this section, we provide more ablations of our OAP and AOP to provide further insight into the functioning of our proposed model. Unless otherwise stated, all OAP ablations are performed on Something-Else compositional split using S3D while AOP ablations are performed on EPIC100-OV using ViT-B/16 as the CLIP image encoder.

## E.1  Number of guiding augmentations per anchor

In this section, we analyze how an increase in the number of guiding augmentations affects the Top-1 accuracy in Something-Else. Guiding augmentations per anchor consist of the temporal gradient of the anchor and a set of object mixing augmentations. The number of object mixing augmentations is bounded by the frequency of a class *e.g.* in EPIC100-OV, the lowest class frequency is 4, hence a maximum of 3 object mixing augmentations can be generated for an anchor belonging to that class. We report the results in Fig. 1 where 0 guiding augmentations mean using only $\mathcal{L}_{out}$ and 1 means adding only temporal gradient. If the expected number of augmentations exceeds the total instances of a class, we truncate it to the maximum possible. Across all the ablations in this section, $\tau_{in} = 0.07$. We observe that performance improves initially by adding temporal gradients and then object mixing augmentations but saturates after 5.

## E.2  Importance of HOI detector

In Tab. 3, we report the performance of using different crops from the HOI detector. To analyze the effect of HOI crops, a fixed prompt *"a picture of a <object>."* is used without fine-tuning base vocabulary or any temporal modeling. We observe that expanding an object crop by taking its union with the contacting hand improves performance. A union provides more spatial context for the CLIP to operate on, especially when objects are occluded by hand or other objects. If more than one crop is generated per frame, we average their features and normalize it to the unit norm (denoted as + in Tab. 3). Ensembling object crops with the full image results in best novel performance.

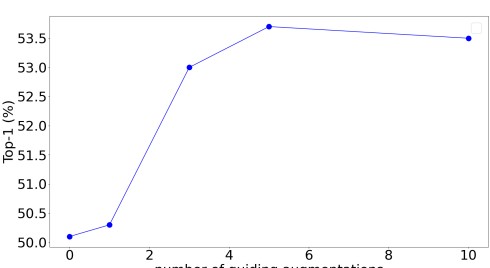

Figure 1: **OAP ablations:** Top-1 acc. vs number of guiding augmentations.

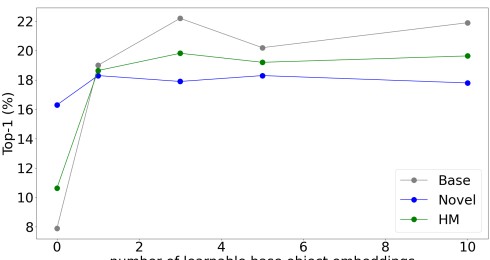

Figure 2: **AOP ablations:** Top-1 acc. vs number of learnable base object embeddings.

### E.3 Learning multiple embeddings for each object class

A single word in CLIP can be divided into multiple tokens, e.g. *"kitchen"* is tokenized separately as *"kitch"* and *"en"*. Similarly, instead of learning a single world embedding for each base object class, we can learn multiple ones. In Fig. 2, we chart the Top-1 performance of using $m$ learnable word embeddings per object class where $m = \{0, 1, 3, 5, 10\}$. $m = 0$ means the base vocabulary is not fine-tuned. We defined the prompt $t_i$ for object $o_i \in \mathcal{O}_{base}$ as $t_i = \{pre_1, ..., pre_{k_1}, W_{o_i}, post_1, ..., post_{k_2}\}$. Here, $W_{o_i} = \{w_{o_i}^1, w_{o_i}^2, ..., w_{o_i}^m\}$ are $m$ learnable word embeddings of $o_i$. In Fig. 2, we observe that both base and novel class performance improves when we fine-tune the vocabulary *i.e.*, $m > 0$. We achieve the best HM for $m = 3$ and the best novel performance for $m = 5$. Again, the fixed prompt *"a picture of a <object>."* is used without any temporal modeling.

Table 3: **HOI crop ablations.** "full" refers to no crop. $\cup$ means object crop expanded with its contacting hand bbox.

| HOI crops | Base | Novel | HM |
|---|---|---|---|
| full | 5.4 | 12.1 | 7.5 |
| objects | 7.9 | 13.5 | 10.0 |
| hands | 5.1 | 10.4 | 6.8 |
| objects $\cup$ hands | **8.6** | 14.4 | **10.8** |
| objects + hands | 6.9 | 14.2 | 9.3 |
| objects $\cup$ hands + full | 7.9 | **16.3** | 10.6 |

Table 4: **Interpretable base object adjustments.**

| original object | nearest neighbor |
|---|---|
| tap | faucet |
| cupboard | compartment |
| pan | saute |
| package | bags |
| meat | brisket |
| fridge | refrigerator |
| egg | eggs |
| Unchanged | plate, bowl, glass, oven, rice |

### E.4 Interpretability analysis of finetuned base vocabulary

For each fine-tuned object class with $m = 1$, we retrieve the discrete token nearest to it in the embedding space. The majority of the fine-tuned vocabulary are not human-readable since one of the primary objectives of fine-tuning was to let the CLIP adjust the class names in a continuous space as opposed to relying on a finite set of discrete tokens. Most of the interpretable adjustments are synonym and plurality corrections (Tab. 4). *"pan"* is adjusted to *"saute"* which reflects the use of pan in EPIC100. We also found that object names *"pizza"* and *"banana"* were adjusted to their corresponding emojis as present in the CLIP vocabulary.

## F Visualization of Guiding Augmentations

We visualize some guiding augmentations in Fig. 3 focusing on certain scenarios. We choose two anchors from Something-Else, *"pickup something"* and *"throw something"* and two from EPIC100, *"cut onion"* and *"open fridge"*. In all cases, the temporal gradients have non-zero motion vectors for static regions due to camera motion. These guiding augmentations can also be regarded as strong [25] or having high variance [27]. Our guiding augmentations can generate a wide range of scenarios two of which are demonstrated via the visualizations.

**Mixed video contains both actions.** We visualize two ordered frames of the object mixed videos for Something-Else in the first two rows, where both actions take place completely. The actions occur simultaneously for *"pickup" i.e.* both figurine and wallet are picked up at the same time while for

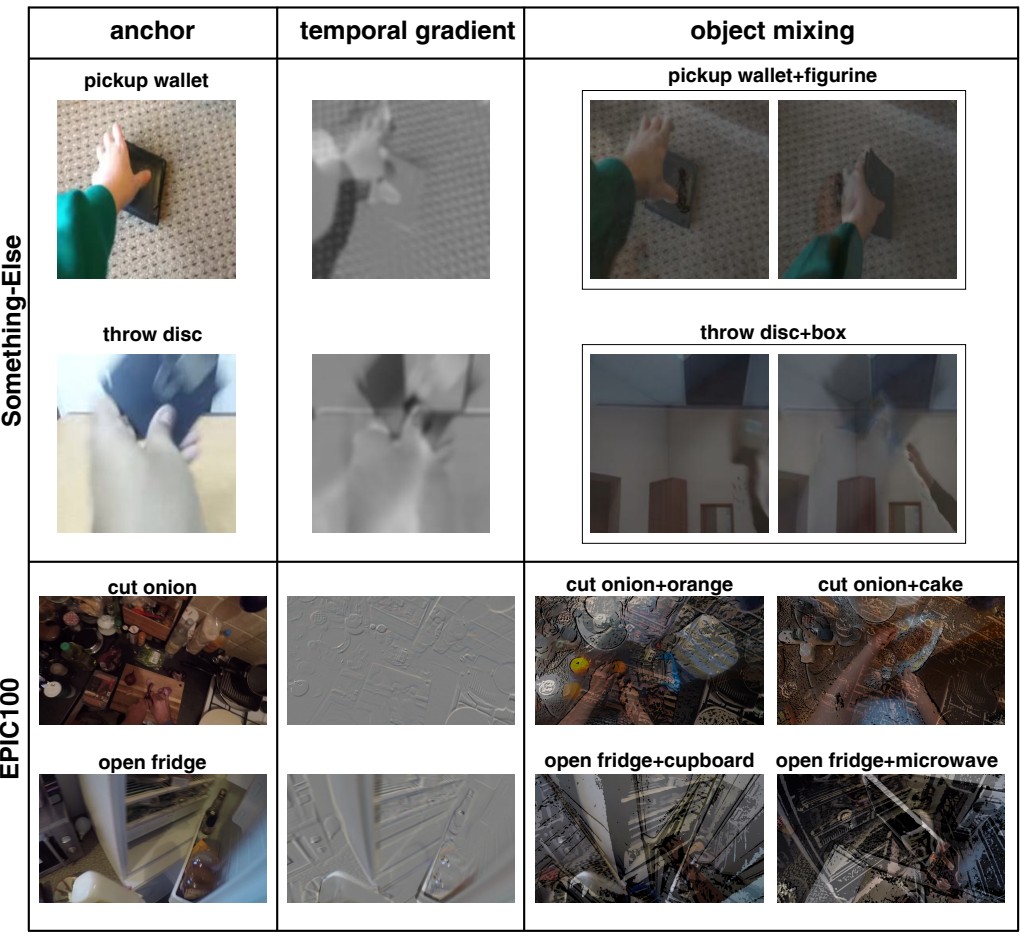

Figure 3: **Visualization of guiding augmentations** for two anchors each from Something-Else and EPIC100 respectively. The object labels per video are not annotated in Something-Else and are provided in the figure for visualization purposes. In EPIC100, the two object mixing augmentations per anchor correspond to a single frame of two different object mixed videos whereas in Something-Else it correspond to two temporally ordered frames of the same object mixed video.

*"throw"* there is a temporal delay between the occurrence of the two *i.e.* first the box is thrown then the disc. Two different objects being thrown simultaneously or successively from different viewpoints in a single video should not change it being recognized as *"throw"*. This provides a free increase of positives per anchor, which benefits rare classes.

**Mixed video contains mixed background.** Next, we visualize guiding augmentations for EPIC100. We purposefully choose anchors having gradients of high noise to observe how the objects are mixed. The noisy temporal gradient introduces noise in the mixed video. It is to be noted that even though the background and its object clutter are mixed, the hand movement of *"cut"* and *"open"* is still temporally preserved. This results in robustness to distractors in the background, which is crucial for generalizing verbs to novel objects.

## G   Broader Impacts

Video understanding research requires extensive compute, which can potentially have a negative impact on the environment, especially when training transformers with a contrastive objective. On the other hand, our proposed active object prompting is parameter-efficient and significantly reduces the computational cost of fine-tuning VLMs. Also, there is potential for video recognition models to be misused especially for unauthorized surveillance.