# OpenReview forum: "Opening the Vocabulary of Egocentric Actions"
_NeurIPS.cc/2023/Conference — NeurIPS 2023 poster_

### Official Review · Reviewer_TsuW · 2023-06-29

**Soundness:** 3 good
**Presentation:** 2 fair
**Contribution:** 3 good
**Rating:** 4
**Confidence:** 5

**Summary:**

In this paper, a new method is proposed for open vocabulary object understanding within egocentric videos for the task of action recognition. The method uses CLIP features to generalise to new objects which haven't been seen during training by focusing on objects using a HoI model, combining nouns via object mixing and creating learned prompts. The method is tested on three datasets, EPIC-Kitchens-100, Assembly 101, and Something-Else. The proposed method outperforms other methods across all datasets on both seen and unseen nouns.

**Strengths:**

Expanding action recognition beyond a closed vocabulary of objects is good to see and the proposed method makes sense to use the CLIP aligned image text space. The results perform well on all three datasets and most aspects of the method are ablated.

**Weaknesses:**

Figure 2 could be improved such that it shows the examples of the anchor, positive, and mixed objects (as well as the negatives). Even with labels, currently, the frames in the figure are quite small and it's difficult to understand what is going on.

Section 4.2: What are the values of k1 and k2 and how were these chosen? How sensitive is the model to these values? In a similar vein, has an investigation between the learnt prompts and what these represent from the learnt embedding space been found?

Line 210: It isn't clear to me what it means by fine-tuning the object embeddings but the CLIP text encoder is kept frozen?

Section 5 the value of alpha and lambda haven't been explored within the paper, I assume these values set to 0.5 have been empirically chosen? If so, how sensitive is the method to these different hyperparameters?





**Questions:**

Here are the list of questions that I would like to see answered, for more context of each question please see the weaknesses section.

1. More information regarding k1 and k2, what are the values and how were they chosen?
2. What do the learnt prompts look like? Do they have a relation to any words that are within the context of the datasets? How much do the learned prompts change per dataset?
3. Could line 210 be clarified as to how the object embeddings are tuned?
4. How were the values of alpha and lambda chosen and how sensitive is the model to these?

**Limitations:**

Yes, the authors have addressed some limitations in the method such as the assumption that every verb/object pair makes a viable action and that CLIP has a strong object bias.

---

> ### Author Rebuttal · Authors · 2023-08-09
>
> We thank the reviewer for their valuable feedback and acknowledging our task of expanding egocentric action recognition beyond a closed vocabulary of objects.
>
> **W1**
>
> *"Figure 2 could be improved..."* - Thank you for the feedback, we will improve this figure in our revision and add a more detailed one in the supplementary.
>
> **W2 \& Q1**
>
> - *"What are the values of k1 and k2 and how were these chosen?"* - k1 and k2 refers to the number of learnable prefix and postfix context prompts. We ran experiments with a wide range of k1 and k2. The results provided in all tables of the main paper are for k1=k2=16.
> - k1=k2=0 refers to CLIP+HOI baseline. Initially, increasing k1,k2 (4→16) improves both base class and overall (HM) performance, until it is too large (32), at which point the model overfits to the base set and cannot generalize well to novel classes. A good balance is achieved (highest HM) for k1=k2=16. See results in the following table.
>
> | **k1,k2** | **Base** | **Novel** | **HM** |
> |:------:|:--------:|:---------:|:------:|
> |  **0** |    7.9   |    **16.3**   |  10.6  |
> |  **4** |  13.2   |    10.5   |  11.7  |
> |  **8** |   21.4   |    7.6    |  11.2  |
> | **16** |  22.0   |    9.6    |  **13.4**  |
> | **32** | **24.2**   |    4.6    |   7.7  |
>
> For the sake of simplicity we assume k1=k2. To measure model sensitivity to the values of only k1 & k2, we refrain from using other AOP strategies like fine-tuning base vocabulary, verb-conditioned prompting and temporal modeling. ViT-B/16 is used as the CLIP-Image encoder backbone. We stop at k1=k2=32 since the maximum context length of CLIP is 77.
>
> **W2 \& Q2**
>
> *"In a similar vein, has an investigation between the learnt prompts"* - While retrieving the learnt prompts, we observed words like *video* and *table*. However, since these prompts are learnt from scratch, most of them are not human-readable when retrieved as discrete tokens. As a result, it is difficult to verify how they behave with the changing of datasets. This is consistent with findings in previous literature [25, a].  Alternatively, we did analyze the interpretability of the fine-tuned base object embedding; this is given in Supplementary `Sec. E.4` which shows base object adjustments like synonym and plurality corrections.
>
> **W3 \& Q3**
>
> *"Could line 210 be clarified as to how the object embeddings are tuned?"* - Similar to learning context prompts by backpropagating the task-specific loss while keeping the CLIP-encoder frozen, we can also do the same for the base object embeddings. The main difference is that unlike the context prompts which are randomly initialized from a normal distribution with 0 mean 0.01 and standard deviation (i.e. learnt from scratch), the object embeddings are initialized as pretrained CLIP text embeddings. Thus backpropagating the loss finetunes the object embeddings and allows the CLIP to adjust the object class names in the continuous space that helps differentiate discrete fine-grained concepts like *‘boom*’ vs *‘arm’*. Also refer to `Sec. E.4` of the supplementary for interpretability analysis of the fine-tuned base object vocabulary.
>
> **W4 \& Q4**
>
> *"How were the values of alpha and lambda chosen and how sensitive is the model to these?"* - $\alpha$ is chosen to be 0.5 and $\lambda$ is chosen to be 1 for all the experiments across all the datasets. We did a hyperparameter sweep for $\lambda = \{0.1, 1, 10\}$ with best performance for $\lambda=1$ where the range of $L_{out}$ and $L_{in}$ were equally scaled. We ran some preliminary experiments with $\alpha=0.2$, however the results were poor hence we aborted the run. Due to compute constraints for end-to-end pretraining of video backbones on large datasets like Something-Else, EPIC and Assembly, we did not perform a full grid-search on $\alpha$. We found $\alpha=0.5$ to be a good estimate since it mixes both the videos with the same weight. We have focused our compute on ablating the main methodological contributions (Sec. E of Supplementary) so that our results are interpretable and reproducible. Please refer to `Sec. B` of the supplementary for additional implementation details.
>
> [a] B. Lester et al., “The Power of Scale for Parameter-Efficient Prompt Tuning”, EMNLP 2021

---

> > ### Comment · Reviewer_TsuW · 2023-08-18
> >
> > Thank you for responding to my initial review. This has cleared up my initial concerns of the paper. After looking at the other reviews and responses I am now leaning towards acceptance for this paper.

---

> > > ### Author Response · Authors · 2023-08-19
> > >
> > > Dear R5 (TsuW),
> > >
> > > Thank you for the kind reply and acknowledging the significance of our work. We are open for further discussion if any extra clarification is needed.
> > >
> > > Best regards,

---

### Official Review · Reviewer_FCUw · 2023-07-03

**Soundness:** 2 fair
**Presentation:** 3 good
**Contribution:** 2 fair
**Rating:** 5
**Confidence:** 3

**Summary:**

The paper tackles open vocabularly action recognition where novel objects may be seen at test time and will be composed with a known verb. Note that the paper does not investigate the open vocabularly of verbs. To address this problem the paper proposes 'object agnostic pretraining' to become less reliant on the objects seen in training as well as 'active object prompting' a method to prompt CLIP for the specific problem of recognizing unseen objects in combination with seen verbs. This approach is tested on compositional action recognition with SomethingElse and open vocabularly action recognition on EPIC-Kitchens-100 and Assembly101.

**Strengths:**

- The set up of open vocabularly benchmarks is thoughtful. For instance, keeping objects from imagenet and kinetics in the base split to avoid leak from pretraining to open vocabularly in test and using tail clases as the open vocabularly ones to make the setting more realistic ensures it will be a useful baseline for future work.

***

- Reporting results on SomethingElse allows comparison to prior work on the related task of compositional action recognition. This is useful as there is no prior work on open vocabularly action recognition to compare to.

***

- The ablations demonstrate the effectiveness of the majority of proposed components

**Weaknesses:**

- Only focusing on the open vocabulary of objects, not verbs, limits the usefulness of the baseline to future works. I appreciate the paper is focused on unseen objects however setting up the baseline in a way that allowed future works to study unseen verbs would have been valuable

***

- The approach is tailored to the specific scenario of egocentric vision and recognizing unseen objects in combination with seen verbs. Testing on SomethingElse, which is not egocentric, did partially alleviate this concern however, the method is still quite specific to the target task.

***

- The baselines in Table 2 are mostly standard action recognition backbones rather than works for compositional action recognition.
- [A] and [B] provide results on SomethingElse and are more recent than [35] so it is unclear why these results weren't compared to.

***

- Limited open vocabulary object recognition baselines in Table 3
- It is unclear why [63] and [18] aren't compared to as the introduction sets these up as some of the most relevant works.
- CLIP performs quite well on novel objects already so makes sense that an existing open vocabulary object detection method built on CLIP would perform even better.

***

- Table 6 ablation, some components ineffective
- From Table 6 the benefit of temporal gradients and L_in seem negligible: +0.2% top-1 accuracy
- Furthermore, it would be useful to assess the impact of these two components separately. For instance, can the object mixing be added without L_in to assess whether L_in is effective or whether the object mixing is the only effective component in this section
- I'm also curious as to whether the success of the object mixing is mainly due to the imbalance of objects in training. How would class-balancing the negative object selected compare to the object mixing.

***

[A] Is an Object-Centric Video Representation Beneficial for Transfer? ACCV 2022.
[B] Revisiting spatio-temporal layouts for compositional action recognition. BMVC 2021.

**Questions:**

Why weren't prior works [A] and [B] on compositional action recognition compared to in Table 1?

Why aren't additional open vocabularly object recognition methods compared to in Table 2? E.g. [63] and [18]

Does L_in have any impact to the final results?

Is object mixing better than class-balancing the object in the (verb) in-class positives?

After reading the rebuttal some of my concerns have been alleviated, namely the comparison with compositional action recognition, the additional ablation, the promise of an open-vocabulary verb baseline in the final version and the clarification on the open vocabulary object detector comparisons. As reading the other reviews has not raised any additional concerns I have upgraded my rating to borderline accept. I have not raised it higher as I am not convinced by the arguments for not considering an open vocabulary of verbs. I do think it is an interesting problem for the paper to focus on an open vocabulary of objects in actions however, the title, abstract and introduction should be updated to clearly indicate this is the focus of the paper if it is accepted. I was also not convinced by the rebuttal response on the proposed object mixing vs. class balancing. An experiment would demonstrate the point more clearly.

**Limitations:**

Limitations are addressed well in a separate section of the main paper.

---

> ### Author Rebuttal · Authors · 2023-08-09
>
> We thank the reviewer for their detailed review, great suggestions and acknowledging the additional experiment on Something-Else. We also thank the author for appreciating the motivation behind our benchmark design. Due to space constraints, kindly find the required results (tables) in the attached rebuttal pdf.
>
> **W1**
>
> We address the concern regarding why an open vocabulary of verbs was not considered in the global author rebuttal. A baseline with unseen verbs is a very nice recommendation. To facilitate future research on both novel verbs and objects, we can release such a benchmark and prepare a zero-shot (verb) + open vocabulary object baseline in the supplementary.
>
> **W2**
>
> - Indeed, our approach is limited to recognizing actions which are defined as a combination of verbs and nouns. However, this type of definition, while prominent in egocentric datasets, is not limited to it and is plausible for any hand-object interaction datasets e.g. Something-Something [16], Charades [a], Breakfast [b] etc.
> - Additionally, contrastive spatiotemporal augmentations for egocentric videos are presently lacking in the community and working with them provides an opportunity for such exploration. Please refer to Sec. C of the supplementary for a detailed explanation.
>
> **W3 \& Q1**
>
> - *"Why weren't prior works [A] and [B]..."* - Thanks for bringing this to our notice! Our OAP is a general pretraining strategy for any video backbone without using any extra parameterized module. Our goal in `Tab. 2` was to show how OAP scales with larger backbones (S3D → Motionformer).
> - [A] uses Motionformer as backbone and [B] uses R3D as backbone on top of which they add a parameterized module specialized for compositional recognition (like STIN [35]). The design of such specialized modules is orthogonal to our approach. In theory, specialized modules on top of our OAP pretrained backbone, should improve the overall performance. Unfortunately [A] do not release their code but in `Tab. 1 of rebuttal pdf` we show this by using CACNF [B] on top of supervised S3D features and then on top of OAP pretrained features. We observe that OAP improves performance since it provides better video features to CACNF. However, the margin of improvement decreases due to CACNF’s inherent capabilities of better compositional recognition. Not that we use S3D with 16 frame input instead of the heavier R3D with 32 frame input as reported in [B].
>
> **W4 and Q2**
>
> As mentioned and visualized `L135` onwards, these SOTA open vocabulary object detectors fail to detect active objects which correspond to the nouns of the action labels. We provide comparison with DETIC which is a better baseline than [63, 18] in open vocabulary object detection. Our method outperforms DETIC by *18.8%*. We could also prepare baselines for [63] and [18] and include it in our revision. Kindly note that naively using CLIP results in poor performance and our proposed AOP-Novel (w/o temporal modeling) finds a good balance between improving both base and novel class performances. Please refer to `Tab. 2 of the rebuttal pdf` for the corresponding comparison.
>
> **W5 and Q3**
>
> - *"...the benefit of temporal gradients..."* - With this ablation we show that how different augmentation affects the Top-1 performance and indeed using temporal gradients does not result in much improvement in accuracy. However, our proposed object mixing strategy brings an improvement of *3.4%* Top-1 Acc.
>
> - *"Does L_in have any impact to the final results?"* - Thanks for pointing this out and this is indeed an important experiment to assess the performance of L_in. In `Tab. 3 of rebuttal pdf`, it can be observed that although object mixing augmentations are extremely effective, using L_in provides a performance boost of *1.6%* Top-1 Accuracy compared to when all the augmentations are used along only with L_out. Here $|G_i| = 5$ and $\tau_{in} = 0.1$.
>
> **W5 and Q4**
>
> - *"Is object mixing better than class-balancing..."* - This is indeed an interesting observation. However, class-balancing the object classes for verb (in-class) positives will require object labels per video which is not provided for Something-Else.
> Secondly, if we assume the object labels are available e.g. EPIC100-OV, Assembly101-OV, our goal of introducing object mixing augmentations and broadly OAP is to remove the dependency of the model on the object for verb recognition. Dependency of object for verb recognition can be introduced in a video model broadly by two ways:
>     - **class imbalance of the dataset:** Some verb-object compositions are seen more frequently than others during training.. Hence, a supervised training with no class-frequency based weighting/sampling will bias the model towards the head compositions. Here, class-balancing might improve performance on the few-shot classes.
>     - **static bias of video models:** A class-balanced model will still be biased towards certain object appearances for verb recognition e.g. assuming *“slicing apple”* is the only action with apple seen during training, during inference if the action *“washing apple”* is shown to the model, the model has a high chance of incorrectly predicting the verb as  *“slicing”* by overfitting to the red and round appearance of the apple. This is widely explored in literature as static bias of video models [7, 28]. Since our motivation is not improving few-shot performance but providing a reasonable solution to the open vocabulary setting, we require the verb model to predict the verb in a novel action composition. Here OAP helps by reducing the dependency of not only the appearance of the object being manipulated but also any distractor objects present in the background (scene).
>
>
> [a] G. Sigurdsson et al., “Hollywood in Homes: Crowdsourcing Data Collection for Activity Understanding”, ECCV 2016
>
> [b] H. Kuehne et al., “The Language of Actions: Recovering the Syntax and Semantics of Goal-Directed Human Activities”, CVPR 2014

---

> > ### Comment · Reviewer_FCUw · 2023-08-17
> > **Response to Rebuttal**
> >
> > Thank you for responding to my initial review. I have no further questions and will enter the review discussion more positive than my original rating.

---

> > > ### Author Response · Authors · 2023-08-19
> > >
> > > Dear R4 (FCUw),
> > >
> > > Thank you for your detailed review, constructive feedback and acknowledging the significance of our work. As the author-reviewer discussion period will end soon, we would love to hear if you have any further comments that facilitate this discussion period or an acceptance decision.
> > >
> > > Best regards,

---

### Official Review · Reviewer_DGnw · 2023-07-06

**Soundness:** 3 good
**Presentation:** 2 fair
**Contribution:** 3 good
**Rating:** 5
**Confidence:** 3

**Summary:**

The paper proposes a novel open vocabulary action recognition task, and this task aims to generalize the ability of the model to an open vocabulary of actions with seen and novel objects. To do this, the authors propose a new method, which mainly contains two components: Object Agnostic Pretraining (OAP) and Active Object Prompting (AOP). The authors also create open vocabulary benchmarks on the EPIC-KITCHENS-100 and Assembly101 datasets, and the results on these benchmarks demonstrate the effectiveness of this method.

**Strengths:**

1) The paper proposes a novel open vocabulary action recognition task, which is important in practice.
2) The method presented in the paper is simple and straightforward.
3) The paper presents sufficient experiments. The results of comprehensive experiments enhance the validity and reliability of the proposed approach.

**Weaknesses:**

1) Motivation is unclear, and many modules are designed without proper motivation, e.g., why conduct object agnostic pretraining, why interested in recognizing active objects.
2) The authors claim that we are the first to explore end-to-end contrastive learning on the video backbone of egocentric videos, but contrastive learning is already widely employed in the fields of video and image.
3) Since this paper focuses on open vocabulary action recognition, why can't the verb category be open vocabulary as well?
4) The proposed methods lack novelty. Firstly, mixup operations are widely used in image augmentation [1]. Secondly, the idea of Parameter-efficient training for Active Object Prompting appears to resemble a modified version of COCOOP [2]. Thirdly, Active Object Prompting (AOP) is not very novel, just a simple combination of the HOI model and CLIP. CLIP has been widely used in open-vocabulary tasks.
5) The description of the paper's method is insufficiently detailed, making it difficult to reproduce. For example, in Line 205, the author mentions ΦL as a projection network without further interpretation or elaboration.

[1] Not All Relations are Equal: Mining Informative Labels for Scene Graph Generation, CVPR2022
[2] Conditional Prompt Learning for Vision-Language Models, CVPR 2022

**Questions:**

It seems that only the objects in this task can be open-vocabulary. Can the verbs in the task be also open-vocabulary?


**Limitations:**

The authors have discussed limitations in chapter 5.6, such as, bias in CLIP.

---

> ### Author Rebuttal · Authors · 2023-08-09
>
> We thank the reviewer for their thoughtful comments and acknowledging the importance of our proposed open vocabulary action recognition task, the simplicity of the proposed approach and sufficient experimentation.
>
> **W1**
>
> - *"why interested in recognizing active objects"* - Our task is action recognition and active objects correspond to the noun in the action labels of our datasets, therefore we want to recognize these objects (please refer to `L54-L63` and `Sec. 4 - Object Encoder`).
> - *"why conduct object agnostic pretraining"* - Our focus is open vocabulary action recognition where actions are composed of a seen verb with a seen or novel object. Video models trained with a set of seen verb-object compositions get biased towards the seen compositions which hampers the generalization ability of the model (please refer to `L42-L53` and `Sec. 4 - Verb Encoder`) i.e. generalizing verbs to novel objects (as can be seen from our experiments in `Tab. 4`). Hence, we conduct object agnostic pretraining (OAP) to remove the dependency of objects on predicting a verb category thus improving its generalization to novel object categories not seen during training.
>
> **W2**
>
> Thanks for noticing! We claim on `L51-L52` that *“To our knowledge, we are the first to explore contrastive learning of video backbones end-to-end on egocentric videos.”* End-to-end (backbone not frozen) contrastive pre-training has been widely explored for Kinetics-style videos [6, 19]. Very recently, it has also been adopted in egocentric videos [30] but for video-text representation learning. On the other hand, some methods only learn a parameterized module over a frozen backbone using contrastive learning [a]. Since egocentric videos in EPIC, Assembly101 and non-ego videos of Something-Else have properties different from Kinetics-style videos (`L224-L226`), the spatio-temporal augmentations needed to train a video backbone end-to-end need to be adapted. We study the nature of such augmentations in `Sec. C` of the supplementary and in this regard, to our knowledge, we are the first to do so.
>
> **W3 \& Question**
>
> *"...why can't the verb category be open vocabulary as well?"* - We address the concern regarding why an open vocabulary of verbs was not considered in the global author rebuttal.
>
> **W4**
>
> - *"Firstly, mixup operations are widely used..."* - [1] uses mixup in the embedding space as opposed to our object mixing which is applied in the pixel space. Kindly note that this is the first time lightweight mixing augmentations of this form are being used for egocentric videos (featuring fine-grained hand-object interactions in the real world).
> - *"Thirdly, Active Object Prompting (AOP) is not very novel..."* - Our goal is to show what design choices are important for prompting a CLIP model to recognize active objects. In this regard, we have found that using an HOI detector and using the union of hand and object crop improves the performance over a base CLIP operating on full image by 3.1 %. Conditional Prompting of a CoCoOp [2] style is another facet of active object prompting but unlike CoCoOp that projects CLIP-Image encoder feature to the text embedding space, we project the verb features learnt via OAP to the text embedding space. This requires no extra computation since the verb encoder is already being used for verb prediction. The most important aspect of AOP which brings about the biggest improvement is *fine-tuning the base object vocabulary*. This is extremely crucial for fine-grained action datasets like EPIC, Assembly101 (`L207-L216`).
> - Additionally,  we are the first to propose open vocabulary action recognition which has many potential applications and might encourage the action recognition community to explore beyond predefined action classes used during training.
>
> **W5**
>
> Thanks for pointing this out! We follow the established CoCoOP [66] meta-network design for $\Phi_L$ i.e. a 2-layer bottleneck MLP (Linear-ReLU-Linear) where the hidden dimension is (input dimension)/16. We will add this information in our supplementary and also release our source code.
>
> [1] A. Goel et al., "Not All Relations are Equal: Mining Informative Labels for Scene Graph Generation", CVPR 2022
>
> [a] D. Singhania et al., “Iterative Contrast-Classify For Semi-supervised Temporal Action Segmentation”, AAAI 2022

---

> > ### Comment · Reviewer_DGnw · 2023-08-17
> >
> > I appreciate the authors' efforts to respond to my points, which have addressed most of my concerns. I would like to keep my previous score.

---

> > > ### Author Response · Authors · 2023-08-19
> > >
> > > Dear R3 (DGnw),
> > >
> > > Thank you for the kind reply and acknowledging the significance of our work. As the author-reviewer discussion period will end soon, we would love to hear if you have any further comments that facilitate this discussion period or an acceptance decision.
> > >
> > > Best regards,

---

### Official Review · Reviewer_KvE6 · 2023-07-06

**Soundness:** 2 fair
**Presentation:** 3 good
**Contribution:** 3 good
**Rating:** 5
**Confidence:** 4

**Summary:**

Existing approaches for egocentric action recognition considers the problem as recognizing actions from a closed set. The manuscript instead proposes to consider this as an open set recognition problem where some objects are unseen during training. Towards, a novel benchmark for evaluating open set egocentric action recognition is proposed using two egocentric action recognition benchmarks. The manuscript also proposes an approach for open set egocentric action recognition. The key idea behind the approach is to decouple the verb representation from interacting objects followed by using a prompt based object encoder. The proposed approach shows strong improvements in the proposed benchmark when compared against existing closed set action recognition baselines.


**Strengths:**

The open set egocentric recognition problem addressed in the manuscript is novel, non trivial and highly relevant to the research community. The development of the two benchmark datasets will be highly beneficial for future research.

The paper is written in a clear and concise manner making it easy to understand the motivations and the proposed approach. An extensive set of ablations are also presented in the manuscript making it clear the impact of the various contributions. Evaluation of the proposed approach on the benchmark and comparison against the existing close set recognition approaches shows its effectiveness in addressing open set egocentric action recognition.


**Weaknesses:**

The main weakness of the manuscript is that the proposed approach is compared only against simple baselines based on standard action recognition approaches. The proposed approach makes use of the CLIP model for object recognition. CLIP is widely known for its performance on zero shot and few shot object recognition. One important baseline would be to use the S3D model for verb recognition (without the proposed OAP training) and the CLIP model for object recognition. The CLIP model can also be used for both verb and object recognition. Without this, it is not clear how effective the proposed approach is for open set egocentric action recognition. Similarly since the novel objects are composed of the tail classes from the respective datasets, using CLIP model would be highly effective since it is good at recognizing few shot and zero shot classes. So the comparisons reported in the paper do not reveal the true impact of the proposed approach.

Some of the design choices behind the proposed approach are not motivated well or empirically proven. For example, why use pre and post context prompts? Why not add them as prefix or suffix to the fixed prompt? For AOP, the verb features are added to the context tokens. Why is this preferable over concatenating the verb features to the input embeddings of the text encoder?

After the OAP pretraining, the verb encoder is further finetuned for supervised verb recognition. Wouldn’t this result in the model losing its object agnostic verb prediction capabilities?

Even though the manuscript presents various ablation studies to validate the effectiveness of various contributions, the experiments are conducted on different datasets. This makes it difficult to analyze the impact of the different contributions of the proposed approach. The reviewer recommends the authors to select a single dataset to present in the ablation analysis.


**Questions:**

Please see the weaknesses section for the main questions regarding the manuscript.

The main problem that the proposed approach is trying to address is related to the fact that existing approaches consider that there is only a fixed set of actions (verb-noun pairs that are observed in the training set). Instead the proposed approach is trying to develop an approach for open set recognition (some object classes are not observed during training). However, the AOP based object encoder still functions in a way that the set of possible objects are already known since the object names are required for the fixed prompt. This contradicts the main objective of the paper to develop an approach for open vocabulary egocentric action recognition. A true open set recognition approach requires a generative model instead of the discriminative one used in the proposed approach.


**Limitations:**

One of the limitations of the proposed approach is the fact all verb-noun pairs cannot constitute a plausible action. This is discussed in the limitations section.

---

> ### Author Rebuttal · Authors · 2023-08-09
>
> We thank the reviewer for their valuable feedback and acknowledging our novel task, benchmarks and the set of ablations.
>
> **W1**
>
> - In the main paper, we have provided the results for *“One important baseline would be...”* The 1st row of `Tab. 4` provides S3D (without OAP) results on EPIC100-OV. The 2nd and 3rd row of `Tab. 3` provide the CLIP-only baseline for active object recognition. Please refer to `Sec. 5.3` for a detailed description of the CLIP-only baseline.
> - *“The CLIP model can also be used...”* - We ran preliminary experiments with CLIP for verb recognition. However due to CLIP’s inherent object bias [42] and lack of temporal knowledge, we omitted these results from our paper as it would not be a fair comparison. Additionally, our verb set is neither open vocabulary nor zero-shot, therefore existing baselines already outperform CLIP. We are happy to add this result in our supplementary.
> - *“Similarly since the novel objects are ….”* - We thank the reviewer for acknowledging our motivation for using CLIP. However, as observed in `Tab. 5`, naively using CLIP doesn't result in good performance, hence we propose AOP to improve CLIP’s performance on both base and novel object classes. Furthermore, we also provide comparisons with DETIC [68] (a SoTA open vocabulary object detector) and C4 [25] (a SoTA zero-shot verb recognition model), both of which are based on CLIP. We outperform DETIC by *26.8%* and C4 by *18.4%*.
>
> **W2**
>
> - *"For example, why use pre and post context prompts?"* - We use pre and post context prompts because it can be learned directly via backpropagation rather than manually engineering different versions of fixed prompts [24, 46] which is laborious and requires domain expertise.
> - *"Why not add them as prefix or suffix to the fixed prompt?"* - The pre and post context prompts are added as prefix and suffix respectively to the fixed prompt where the content of the fixed prompt might change. Since our purpose is to learn the prefix and postfix context prompts, keeping the fixed prompt simple results in better performance. In the following table, we provide the comparison between the two prompt versions which empirically supports our claim. Other AOP components are restricted.
>
> | **Prompt**     | **Base** | **Novel** |  **HM**  |
> |----------------|:--------:|:---------:|:--------:|
> | "$W_{pre}, W_{a\ photo \ of \ a \ <object>.} , W_{post}$"     |   21.4   |    7.2    |   10.8   |
> |  "$W_{pre}, W_{<object>} , W_{post}$" | **22.0** |  **9.6**  | **13.4** |
>
> - *"For AOP, the verb features are added to the context tokens."* - If we understood correctly, the reviewer asks if we add the verb features to the context tokens. The verb features are indeed added to the embeddings of the context tokens. We empirically found that addition is better than concatenation although the difference in performance is negligible. Also it is a standard practice in conditional prompting [66, 67]. In the following table, we provide concatenation vs addition results, where for concatenation, the verb feature from S3D (2048-D) is first concatenated to the context prompt embedding (512-D for CLIP-ViT-B/16) and then the projection network $\Phi_L$ is applied to project it back to 512-D. Other AOP components are restricted.
>
> | **Method**    | **Base** | **Novel** |  **HM**  |
> |---------------|:--------:|:---------:|:--------:|
> | concatenation |   24.9   |    13.1   |   17.2   |
> | addition      | **25.4** |  **13.2** | **17.4** |
>
> **W3**
>
> We thank the reviewer for raising this important question. The model does not lose its object agnostic capabilities when fine-tuned for supervised verb recognition as is evident from our experiments and ablations. To ensure this, we adopt the following fine-tuning strategy. After OAP pretraining, a linear classification layer is added to the encoder whose weights are initially randomly sampled from a normal distribution with 0 mean and 0.01 standard deviation. Now during fine-tuning, if the classification layer is trained with learning rate L, then the base network (which has OAP pretrained weights) is fine-tuned with a learning rate of L/10. This ensures that object-agnostic verb representations are not forgotten. The learning rate L used for fine-tuning itself is very low (1e-3). We find that fine-tuning using the above strategy outperforms linear probing the object-agnostic verb features. This is also a standard practice in UberNCE [20].
>
> **W4**
>
> We thank the reviewer for this suggestion. In an ideal setting, we would like to be able to use the same dataset for all ablation studies. However, Something-Else does not have object classes annotated per video as stated in `L228-L229`. We perform all OAP ablations on Something-Else and all AOP ablations on EPIC-OV split. We purposely choose Something-Else for OAP ablations since it already has existing compositional baselines which we can compare to. For AOP, we choose EPIC-KITCHENS since the HOI detector [52] is already trained on a subset of it and is less likely to confound the ablation results.
>
> **Questions:**
>
> We think there might be a confusion here between open-set recognition versus open-vocabulary recognition. Indeed, the former would require a generative model instead of a discriminative one. However, in an open vocabulary setting, it is assumed that during inference, the class names are provided and there is no prior assumption as such on the vocabulary of the names provided during inference, hence *open vocabulary*. This is a standard practice followed by other open-vocabulary works [18, 68]. We make this distinction in `L30-31`, but will be sure to further highlight the differences in our revision.

---

> > ### Comment · Reviewer_KvE6 · 2023-08-17
> > **Response to authors**
> >
> > I thank the authors for the rebuttal which addressed most of my concerns. As a result, I change my rating to borderline accept.

---

> > > ### Author Response · Authors · 2023-08-19
> > >
> > > Dear R2 (KvE6),
> > >
> > > Thank you for the kind reply and acknowledging the significance of our work. As the author-reviewer discussion period will end soon, we would love to hear if you have any further comments that facilitate this discussion period or an acceptance decision.
> > >
> > > Best regards,

---

### Official Review · Reviewer_jmBw · 2023-07-08

**Soundness:** 3 good
**Presentation:** 3 good
**Contribution:** 3 good
**Rating:** 5
**Confidence:** 3

**Summary:**

Learning representations that generalise across novel objects and actions is an important topic of research that is of great practical utility. This is especially true due to most datasets containing only a few combinations of object-action pairs or triplets, which often leads to model overfitting. A key question is, how to overcome this dataset limitation. The approach tackles this by decoupling the object representation and verb representation by using object word embeddings and verb-agnostic visual representations.

- Task: Given objects and actions seen during training, generalize to new actions that are performed with either seen or novel objects.
- Approach: Decompose predictions of objects and verbs. This overcomes the verb-object overfitting that might cause lack of generalization to novel verb-object combinations.
    - Object-agnostic pretraining of a verb encoder
    - Prompt-based object encoder (CLIP features for a given prompt). This enables an open vocabulary (of objects that CLIP is trained with).
- Contributions:
    - Open vocab. benchmarks on EPIC-KITCHENS-100, Assembly101
    - Approach that decomposes object and verb encodings is better than prior approaches at recognizing novel verb-object combinations.

**Strengths:**

1. Decoupling the object representation into a function of (a) object word embedding, and (b) object-agnostic verb embedding is a promising formulation. While the specific details of the verb-agnostic encoder training algorithm and prompting of the text encoder might change and improve, the basic formulation that decouples the two different semantic concepts, makes sense. This  leverages CLIP’s pretrained object-text representation that is trained over large datasets containing many variations of objects.
2. CLIP text encoder’s input “context prompt” into a function of (noun CLIP word embedding, object-agnostic verb encoding ) is interesting. The paper mentions that the “pre” and “post” are necessary for the encoding to work. Is this a novel observation or is there prior literature on the topic?
3. In addition to the verb encoder, the paper aims to more accurately represent the differences between fine-grained objects via CLIP’s text encoder. To achieve this, the approach also involves fine-tuning the object embeddings (on seen object classes). The paper demonstrates that this achieves a significant improvement in performance.

**Weaknesses:**

1. There are many small decision choices involved in the successful implementation of the approach. There are different types of augmentation and contrastive sampling. Further, there are different representations for nouns, verbs. The noun object representation is also fine-tuned. It makes the overall approach appear to be the sum of a series of smaller tricks. While it makes the paper a bit harder to read and understand, from a practitioner’s standpoint, this might indeed contribute to new knowledge / understanding of what’s possible. Further, it’s nice that Table 5 evaluates the effects of these different decision choices.
2. The visual feature extraction heavily relies on a HOI detector to obtain the “active object”. This only works on hand-based interactions. This is a fairly narrow set of actions compared to overall possible object-verb interactions. However, one can imagine that this can also be extended to object detectors and relationship-detectors in non-ego-centric RGB videos. Yet again, the approach would have to rely on the high performance of these detectors. A concrete question along these lines is: how good is the HOI detector? What’s the UB performance of the proposed approach assuming perfect HOI detection?
3. There’s an assumption that non-static ⇒ active objects ⇒ objects involved in action. And vice-versa, i.e., static objects are *not* of interest. The masking (or not) of objects is based on this assumption? Is this a valid assumption? I raise this point because the validity of this assumption requires some discussion, and the implementation of this assumption (masking), is also understandably imperfect, and results in noisy augmentations.
4. To produce a training set containing diverse objects for the same verb, the approach does data augmentation by masking out certain objects, and alpha-blending other objects in. This results in unrealistic, noisy images. The MIL-NCE-like loss appears to counter this effectively. However, is it possible to perform better augmentation like [54, 55]? How much will performance improve with better augmentation?
5. Limitations: The approach is only as “open” as the vocabulary of objects that CLIP’s feature space encodes. It is also heavily dependent on the performance of the HOI detector.

**Questions:**

**Clarity / Details**
    1. Fig. 4 and text don’t go well together, or I’m confused. f_\theta = f_v and g_theta (f_theta(.)) = f_v^L ?
    2. In Fig. 4, the verb encoder is shown to be frozen. This confuses me. My understanding was that the training procedure in Sec. 4.1 indeed updates the parameters of the verb encoder.
    3. The input to the verb encoder is the whole video. Is it the same for the HOI detector? Then how are image crops sampled from this video (to pass through CLIP Image Encoder)?

**Limitations:**

Yes, they discuss limitations.

---

> ### Author Rebuttal · Authors · 2023-08-09
>
> We thank the reviewer for the thoughtful comments and acknowledging our open vocabulary benchmark, our method’s potential and our ablations in Table 5.
>
> **S1**
>
>  *“This paper mentions that the “pre” and “post” are….”* - Using learnable prefix and postfix context prompts [13, 25, 66] are used as a (parameter-efficient) alternative to manually engineering prompts [24, 46] which is laborious and requires domain expertise. Some papers like ViLD [18] use an ensemble of 63+ prompt templates to report their final result.
>
> **W1**
>
> In our approach, the main contributions are:
> - **object mixing augmentations** to generate object-agnostic verb representations
> - **use of L_in** to mitigate the noise in object mixing augmentations introduced as a result of the persistent camera motion in egocentric videos. In
> - **verb-conditioned** prompt generation to provide additional context to CLIP when the active object is ambiguous e.g. for the action of “beating egg”, the objects being interacted with are whisk and bowl, but the active object is egg.
> - **fine-tuning** base object vocabulary which allows CLIP to adjust the object class names in a continuous space that helps differentiate discrete fine-grained concepts like ‘boom’ vs ‘arm’.
>
> **W2**
> - *"The visual feature extraction heavily relies..."* - In presently available large-scale egocentric datasets like Ego4D, EPIC-KITCHENS and Assembly101, all annotated actions are hand-object interactions where the motion of the hand is described by a verb and the object by a noun. One advantage of our pipeline is that it is sufficiently general and individual components can be replaced with a better one in the future. In this essence, our HOI detector can be replaced with object/relationship -detectors.
> - *"Yet again, the approach would have to rely on..."* - We acknowledge that using an HOI detector contributes to our success as can be observed from Tab. 5 (+3.1%). However, its baseline performance is still poor and further ablations demonstrate that design strategies such as fine-tuning base object vocabulary (+8.6%) and verb-conditioned prompting (+3.9%) are extremely beneficial for AOP.
> - *"A concrete question along these lines is..."* - Thank you for this great suggestion. However, the HOI detector [52] cannot be evaluated directly on EPIC100 nor Assembly101 because these datasets do not have ground truth bounding box ground annotations. The reason for choosing [52] as our HOI detector is because of its strong cross-dataset generalization (at worst, it obtains 92.9% of the mAP of training and testing on the same dataset). Also, [52] is trained on a subset of EPIC and hence will not confound the AOP ablation study on EPIC100.
>
> **W3**
>
> We don’t assume that static objects are not of interest. Rather, we assume that actions are related to non-static regions that we refer to as active regions in the paper `(L167-L168)`. Active regions may still contain static objects, e.g. for the action of “cutting onions”, the cutting board itself is static but relevant for the action. Our use of masking aims to minimize the impact of distractor objects outside of the active region.
>
> **W4**
>
> We thank the reviewer for this suggestion. We expect expensive handcrafted augmentations, especially NeuralDiff [54] to improve performance as it can mitigate the impact of camera motion. However they cannot be generated online during mini-batch training and need to be pre-computed offline. Generating static views from egocentric videos is a research topic in itself [54] and is beyond the scope of our work.
>
> **W5**
>
> - CLIP’s feature space encodes a variety of “concepts” which are not necessarily objects but certainly have a bias toward them i.e. low-level properties like texture, shape, color, etc. to high-level properties like scene and context. Exploring the limitations of CLIP [a, 42] and addressing them to perform better compositional reasoning [b], verb reasoning [25, 57], etc. is a broad research topic. Our main aim is to recognize an open vocabulary of active objects using a VLM like CLIP and we are the first to do so. However, our entire pipeline and training strategies are general where individual components like CLIP can be replaced with a better VLM if it is proposed in the future.
> - Moreover, targeting open vocabulary using CLIP is already established in object detection [18, 68] and our proposed AOP outperforms all these baselines as shown in Tab. 3.
>
> **Questions:**
>
> Fig.4 shows training/testing protocol for AOP. The verb encoder, already trained using OAP `(Sec. 4.1)` is kept frozen during AOP `(Sec. 4.2)` to prevent the object-agnostic verb features from getting corrupted. HOI detector takes each frame of the video as input. Each frame is then cropped on the basis of the detected hand-object interaction and then passed to the CLIP image encoder. Further details for the HOI detector are provided in `Sec. B` and `Sec. E.2` of the supplementary.
>
> [a] M. Lewis et al., “Does CLIP Bind Concepts? Probing Compositionality in Large Image Models”, arXiv 2022
>
> [b] N. Nayak et al., “Learning to Compose Soft Prompts for Compositional Zero-shot Learning”, ICLR 2023

---

### Author Rebuttal · Authors · 2023-08-09

We thank all the reviewers for their valuable time and constructive feedback. We also thank them for appreciating our novel problem formulation of open vocabulary action recognition.

We appreciate the concern from R3 (DGnw) and R4 (FCUw) for not considering an open vocabulary of verbs; here we explain our rationale for keeping a closed verb set:

- **A closed set is logical, semantically:** Verbs are characterized by the hand movements when manipulating an object. Yet fine-grained hand motions are inherently bounded to a limited set of categories. This is confirmed by haptics research, which define only 17 [a] to 34 [b] different hand grasps.  Furthermore, only 5-10 of these grasps are in use on a regular basis [b].

- **Limitations of Datasets:** Existing datasets do not lend themselves well to investigate open verb vocabularies.  Verbs in datasets of interest (EPIC-KITCHENS-100 [8], Assembly101 [51], Ego4D [17]), are limited in number (24 in Assembly, ~100 for EPIC100 and Ego4D).  They are also inherently redundant in the underlying movements, e.g. consider the same hand motion from *screwing a wheel*, *opening a bottle cap*, and *juicing a lemon*.  As such, any partitioning is still likely to have information leakage from the closed to the open set.

In conclusion, we appreciate the idea of open vocabulary verbs.  We leave it for future work, where the topic can be addressed with due consideration, in a more meaningful setting and with better data.  Furthermore, we wish to emphasize that using a closed verb set does not diminish our current contribution - we are the first to open the vocabulary of objects for hand-object interaction, and see our paper as a starting point for investigating open vocabulary action recognition.

[a] T. Feix et al., “The GRASP Taxonomy of Human Grasp Types”, IEEE Transactions on Human-machine Systems 2014

[b] IM Bullock et al., "Grasp Frequency and Usage in Daily Household and Machine Shop Tasks", IEEE Transactions on Haptics 2013

---

### Decision · Program_Chairs · 2023-09-21

**Decision:**

Accept (poster)

**Comment:**

The paper studied open-vocabulary recognition of human-object interactions in egocentric videos. The initial review ratings were mixed. The reviewers raised concerns about the problem formulation (e.g., open vocabulary recognition for nouns but not for verbs) and the empirical evaluation (e.g., lack of comparison to certain baselines). They also questioned several technical details. The authors provided a detailed response. All reviewers were satisfied with the rebuttal, and leaned toward the acceptance of the paper. Reviewer TsuW expressed the support in the discussion, yet did not update the rating.

After going through the paper, the reviewers, and the responses, the AC considers that the paper addressed an interesting problem and presented a reasonable solution, and thus recommend its acceptance, despite an overall borderline rating.